# Learning in Bayesian Stackelberg Games With Unknown Follower's Types

**Matteo Bollini** [1]  **Francesco Bacchiocchi** [1]  **Samuel Coutts** [2]  **Matteo Castiglioni** [1]  **Alberto Marchesi** [1]

## Abstract

We study online learning in *Bayesian Stackelberg games*, where a leader repeatedly interacts with a follower whose *unknown private type* is independently drawn at each round from an *unknown* probability distribution. The goal is to design algorithms that minimize the leader's *regret* with respect to always playing an optimal commitment computed with knowledge of the game. We consider, for the first time to the best of our knowledge, the most realistic case in which the leader does *not* know anything about the follower's types, *i.e.*, the possible follower payoffs. This raises considerable additional challenges compared to the commonly studied case in which the payoffs of follower types are known. First, we prove a strong negative result: no-regret is unattainable under *action feedback*, *i.e.*, when the leader only observes the follower's best response at the end of each round. Thus, we focus on the easier *type feedback* model, where the follower's type is also revealed. In such a setting, we propose a no-regret algorithm that achieves a regret of $\widetilde{O}(\sqrt{T})$, when ignoring the dependence on other parameters.

## 1. Introduction

*Stackelberg games* (SGs) (Von Stackelberg, 1934) are foundational economic models that capture *asymmetric* strategic interactions among rational agents. In an SG, a *leader publicly commits to a strategy* beforehand, and a *follower* then best responds to this commitment. This simple form of interaction underlies several more complex economic models, such as, *e.g.*, *Bayesian* persuasion (Kamenica & Gentzkow, 2011), contracts (Grossman & Hart, 1992), auctions (Myerson, 1981), and security games (Tambe, 2011).

The problem of *learning an optimal strategy to commit to* in

---

SGs has recently received growing attention (see, *e.g.*, (Peng et al., 2019; Fiez et al., 2020; Bai et al., 2021; Lauffer et al., 2023; Balcan et al., 2025b)). In particular, in this paper we study *online learning* in *Bayesian SGs* (BSGs), where a leader repeatedly interacts with a follower over $T$ rounds, with the follower having a (different) *unknown private type* at each round. The follower's type determines the follower's payoffs and, consequently, the best response they play. At each round $t$, the leader commits to a strategy prescribed by a learning algorithm, and the follower then best responds to it. The goal of the leader is to minimize their *(Stackelberg) regret*, which measures how much utility they lose over the $T$ rounds compared to always committing to an optimal strategy in hindsight. Ideally, one would like learning algorithms that are *no-regret*, meaning that their regret grows sublinearly in the number of rounds $T$.

Online learning in BSGs has already been investigated in the literature, both when the follower's types are selected adversarially (Balcan et al., 2015; 2025a) and when they are independently drawn at each round according to some *unknown* probability distribution (Personnat et al., 2025). However, all these works rely on the very stringent assumption that *the leader knows the payoffs associated with every possible follower type*. This assumption is rather unreasonable in practice. For instance, in security games it would amount to assuming that the defender knows the target preferences of every possible malicious attacker profile.

We consider, for the first time to the best of our knowledge, online learning in BSGs where the leader does *not* know anything about the game, including the payoffs of possible follower types and the probability distribution according to which such types are drawn at each round. Our main result is *the first no-regret learning algorithm for BSGs that does not require any assumptions on the leader's knowledge*.

### 1.1. Original Contributions

We start by proving a strong negative result for the *action feedback* model, where, at the end of each round, the leader only observes the best-response action played by the follower. Specifically, we show that there exist BSGs in which any learning algorithm must incur regret that grows exponentially in the number of bits needed to represent follower payoffs. Thus, even in instances where only a few bits are

---

[1]Politecnico di Milano, Milan, Italy [2]Massachusetts Institute of Technology, Cambridge, MA, USA. Correspondence to: Matteo Bollini <matteo.bollini@polimi.it>.

*Proceedings of the 43rd International Conference on Machine Learning*, Seoul, South Korea. PMLR 306, 2026. Copyright 2026 by the author(s).

sufficient to represent follower payoffs, the regret can be prohibitively large.

In the remainder of the paper, we focus on the easier *type feedback* model, where, at the end of each round, the leader not only observes the follower's best response but also their type. In such a setting, we provide a no-regret learning algorithm for the leader that does *not* require any knowledge of follower payoffs in order to operate.

We remark that, under type feedback, the leader does not directly observe the payoffs associated with the follower's type, as the type itself serves only as an identifier associated with those payoffs. This setting naturally arises in applications where the leader can determine whether two previously encountered followers belong to the same type, even without knowing their exact payoffs. For instance, in security games, a guard may observe the type of a criminal they encounter without knowing the criminal's target preferences. Similarly, web platforms may observe the category of a user without directly observing their preferences.

Our no-regret learning algorithm works by splitting the time horizon into *epochs*. At each epoch $h$, the algorithm learns the follower's best-response regions in order to identify the leader's commitments that are at most $\epsilon_h$-suboptimal. Then, in the following epoch $h + 1$, the algorithm restricts the leader's decision space to such commitments and halves the suboptimality level $\epsilon_h$. This allows the algorithm to use commitments that are not overly suboptimal in the next epoch and thus keep the regret under control.

At each epoch $h$, our algorithm performs three main steps:

- First, the algorithm uses $\mathcal{O}(1/\epsilon_h^2)$ rounds of interaction with the follower to build a suitable estimator of the probability distribution over types $\mu$. At the same time, it identifies a subset of types whose probability is at least $\epsilon_h$. This allows it to filter out types that occur with too low probability, for which learning best-response regions would require too many rounds.

- The second main step is to learn the polytopes defining the best-response regions for the follower types identified in the previous step. This is done by adapting a procedure developed by Bacchiocchi et al. (2025) for learning in *non-Bayesian* SGs.

- The third and final step of the epoch is to use the information on best-response regions collected so far to compute a suitable set of $\epsilon_h$-suboptimal commitments to be used in the subsequent epoch $h + 1$. This crucial step ensures that all leader commitments selected at that epoch will be at most $\mathcal{O}(\epsilon_h)$ suboptimal.

Our no-regret algorithm achieves regret of order $\widetilde{\mathcal{O}}(\sqrt{T})$ in $T$ and depends polynomially on the size of the BSG instance

when the number of leader actions $m$ is fixed. Notice that an exponential dependence on $m$ in the regret is unavoidable, as shown by Peng et al. (2019), even in the simpler *non-Bayesian* SGs with a single follower type.

## 1.2. Related Works

Our paper is primarily related to the line of research on online learning in SGs with finite action spaces (*i.e.*, *normal-form* SGs). Specifically, Letchford et al. (2009); Peng et al. (2019); Bacchiocchi et al. (2025) investigate online learning in single-follower, *non-Bayesian* SGs, by bounding the sample complexity of learning an optimal commitment. Personnat et al. (2025) consider the regret-minimization problem in multi-follower BSGs with *stochastic* follower types, while Balcan et al. (2015; 2025a) consider single-follower BSGs with adversarially selected follower types. These works assume knowledge of the payoffs of all the follower types. In contrast, we study, for the first time to the best of our knowledge, regret minimization in BSGs with *unknown* follower payoffs.

Other lines of research study models that are less closely related to ours. Fiez et al. (2020) study SGs with continuous action spaces and analyze gradient-based learning dynamics, establishing equilibrium characterization and convergence guarantees. Bai et al. (2021) consider SGs where both players learn from noisy bandit feedback. Lauffer et al. (2023) study dynamic SGs with a *Markovian* state that affects the leader rewards and available actions.

Finally, our work is also related to online learning in similar settings, such as Bayesian persuasion (Castiglioni et al., 2020; Wu et al., 2022; Bernasconi et al., 2023; Gan et al., 2025; Lin & Li, 2025; Bacchiocchi et al., 2024a) contract design (Ho et al., 2014; Cohen et al., 2022; Zhu et al., 2023; Bacchiocchi et al., 2024b) and security games (Blum et al., 2014; Balcan et al., 2015; Peng et al., 2019).

## 2. Preliminaries

### 2.1. Bayesian Stackelberg Games

A *Bayesian Stackelberg game* (BSG) (Letchford et al., 2009) is characterized by a finite set $\mathcal{A}_L := \{a_i\}_{i=1}^m$ of $m$ leader actions and a finite set of different follower types $\Theta$, with $K := |\Theta|$. W.l.o.g., we assume that all follower types share the same action set of size $n$, denoted by $\mathcal{A}_F := \{a_j\}_{j=1}^n$. The payoffs of the leader are encoded by the utility function $u^L : \mathcal{A}_L \times \mathcal{A}_F \to [0, 1]$, while each follower's type $\theta \in \Theta$ has a payoff function $u_\theta^F : \mathcal{A}_L \times \mathcal{A}_F \to [0, 1]$.

In a BSG, the leader commits in advance to a *mixed strategy* (hereafter simply referred to as a *commitment*), which is a probability distribution $x \in \Delta(\mathcal{A}_L)$ over leader's actions.[1]

---

[1] Given a finite set $X$, we denote by $\Delta(X)$ the set of all the

Each $x_i \in [0, 1]$ denotes the probability of selecting action $a_i \in \mathcal{A}_L$. Then, a follower's type $\theta \in \Theta$ is drawn according to a probability distribution $\mu \in \Delta(\Theta)$, *i.e.*, $\theta \sim \mu$. W.l.o.g., we assume that the follower, after observing the leader's commitment $x \in \Delta(\mathcal{A}_L)$, plays an action deterministically. Specifically, the follower plays a *best response*, which is an action maximizing their expected utility given the commitment $x \in \Delta(\mathcal{A}_L)$. For every follower's type $\theta \in \Theta$, the set of follower's best responses is[2]

$$A_\theta^F(x) := \arg\max_{a_j \in \mathcal{A}_F} \sum_{i \in [m]} x_i u_\theta^F(a_i, a_j).$$

As is customary in the literature, we assume that, when multiple best responses are available, the follower breaks ties in favor of the leader by choosing an action that maximizes the leader's expected utility. Formally, a follower of type $\theta \in \Theta$ selects a best-response action $a_\theta^\star(x) \in A_\theta^F(x)$ such that

$$a_\theta^\star(x) \in \arg\max_{a_j \in A_\theta^F(x)} \sum_{i \in [m]} x_i u^L(a_i, a_j).$$

Throughout the paper, with a slight abuse of notation, given a commitment $x \in \Delta(\mathcal{A}_L)$ and a follower's action $a_j \in \mathcal{A}_F$, we denote $u^L(x, a_j) := \sum_{i \in [m]} x_i u^L(a_i, a_j)$, and, for every type $\theta \in \Theta$, we let $u_\theta^F(x, a_j) := \sum_{i \in [m]} x_i u_\theta^F(a_i, a_j)$.

The leader's goal is to find an *optimal commitment*, which is a mixed strategy $x \in \Delta(\mathcal{A}_L)$ that maximizes the leader's expected utility, assuming that the follower always responds by selecting a best-response action. Formally, the leader faces the optimization problem: $\max_{x \in \Delta(\mathcal{A}_L)} u^L(x)$, where, for ease of notation, we let $u^L(x) := \sum_{\theta \in \Theta} \mu_\theta u^L(x, a_\theta^\star(x))$ be the leader's expected utility under the follower's type distribution $\mu$. Notice that the existence of an optimal commitment $x^\star \in \arg\max_{x \in \Delta(\mathcal{A}_L)} u^L(x)$ is guaranteed by the fact that the follower breaks ties in favor of the leader. We denote by $\text{OPT} := u^L(x^\star)$ the leader's expected utility under an optimal commitment. Furthermore, given a leader's commitment $x \in \Delta(\mathcal{A}_L)$ and a parameter $\epsilon \in (0, 1)$, we say that $x$ is $\epsilon$-optimal if it holds that $u^L(x) \geq \text{OPT} - \epsilon$.

**Follower's Best-Response Regions** For every follower's type $\theta \in \Theta$ and action $a_j \in \mathcal{A}_F$, we define the *best-response region* $\mathcal{P}_\theta(a_j) \subseteq \Delta(\mathcal{A}_L)$ as the set of leader's commitments under which the utility of type $\theta$ is maximized by playing action $a_j$. Formally, this region can be written as

$$\mathcal{P}_\theta(a_j) := \Delta(\mathcal{A}_L) \cap \left( \bigcap_{a_k \in \mathcal{A}_F \setminus \{a_j\}} \mathcal{H}_\theta^{jk} \right),$$

where $\mathcal{H}_\theta^{jk}$ is the half-space in which action $a_j$ is preferred over action $a_k$ by a follower of type $\theta$. Formally:

$$\mathcal{H}_\theta^{jk} := \{x \in \mathbb{R}^m \mid u_\theta^F(x, a_j) \geq u_\theta^F(x, a_k)\}.$$

We observe that the best-response region $\mathcal{P}_\theta(a_j)$ is a polytope, since it is bounded and defined as the intersection of half-spaces. Specifically, it is defined by at most $n - 1$ *separating* half-spaces $\mathcal{H}_\theta^{jk}$ and $m$ *boundary* half-spaces ensuring $x_i \geq 0$ for all $i \in [m]$. Consequently, the number of vertices of each $\mathcal{P}_\theta(a_j)$ is upper bounded by $\binom{n+m}{m}$.

In the following, given a subset of types $\Theta' \subseteq \Theta$, we denote by $\mathcal{A}_F(\Theta')$ the set of *action profiles* $\boldsymbol{a} := (\boldsymbol{a}_\theta)_{\theta \in \Theta'}$ specifying an action $\boldsymbol{a}_\theta \in \mathcal{A}_F$ for each follower's type $\theta \in \Theta'$. Given an action profile $\boldsymbol{a} \in \mathcal{A}_F(\Theta')$, we let

$$\mathcal{P}(\boldsymbol{a}) := \bigcap_{\theta \in \Theta'} \mathcal{P}_\theta(\boldsymbol{a}_\theta) \subseteq \Delta(\mathcal{A}_L)$$

be the polytope of all the leader's commitments under which action $\boldsymbol{a}_\theta$ is a best response for every type $\theta \in \Theta'$.[3] Furthermore, given a profile $\boldsymbol{a} \in \mathcal{A}_F(\Theta)$, we let $\boldsymbol{a}|\Theta' \in \mathcal{A}_F(\Theta')$ be the restriction of $\boldsymbol{a}$ to the types in $\Theta' \subseteq \Theta$; formally, $\boldsymbol{a}|\Theta' := (\boldsymbol{a}_\theta)_{\theta \in \Theta'}$. Throughout the paper, we assume that there exists an optimal commitment $x^\star \in \mathcal{P}(\boldsymbol{a}^\star)$ for some action profile $\boldsymbol{a}^\star \in \mathcal{A}_F(\Theta)$ such that $\text{vol}(\mathcal{P}(\boldsymbol{a}^\star)) > 0$ and $a_\theta^\star(x^\star) = \boldsymbol{a}_\theta^\star$ for every $\theta \in \Theta$.[4]

### 2.2. On the Representation of Numbers

We assume that all numbers manipulated by our algorithms are rational. Rational numbers are represented as fractions, specified by two integers encoding the numerator and the denominator (Schrijver, 1998). Given a rational number $q \in \mathbb{Q}$ represented as a fraction $b/c$ with $b, c \in \mathbb{Z}$, we define the number of bits required to store $q$ in memory, referred to as its *bit-complexity*, as $B_q := B_b + B_c$, where $B_b$ ($B_c$) denotes the number of bits needed to represent the numerator (denominator). For ease of presentation, and with a mild abuse of terminology, given a vector in $\mathbb{Q}^D$ consisting of $D$ rational numbers represented as fractions, we define its bit-complexity as the maximum bit-complexity among its entries. Throughout the paper, we assume that every payoff $u_\theta^F(a_i, a_j)$ and $u^L(a_i, a_j)$ has bit-complexity bounded by some $L \in \mathbb{N}$, which is known to the leader.

### 2.3. Online Learning in Bayesian Stackelberg Games

We study settings in which the leader *repeatedly* interacts with the follower over multiple rounds. We assume that the

---

probability distributions over $X$.

[2]In this paper, we compactly denote by $[b] := \{1, \ldots, b\}$ the set of the first $b \in \mathbb{N}$ natural numbers.

[3]For ease of notation, we omit the dependency on $\Theta' \subseteq \Theta$ in $\mathcal{P}(\boldsymbol{a})$, and we assume that such a dependency is implicitly encoded in the action profile $\boldsymbol{a} \in \mathcal{A}_F(\Theta')$.

[4]Given a subset $\Theta' \subseteq \Theta$ and a tuple $\boldsymbol{a} \in \mathcal{A}(\Theta')$, we denote by $\text{vol}(\mathcal{P}(\boldsymbol{a}))$ the volume (relative to $\Delta(\mathcal{A}_L)$) of the polytope $\mathcal{P}(\boldsymbol{a})$.

leader has no knowledge of either the distribution over types $\mu$ or the utility function $u_\theta^F$ of each type $\theta \in \Theta$.

At each round $t \in [T]$, the leader-follower interaction unfolds as follows:

1. The leader commits to a mixed strategy $x_t \in \Delta(\mathcal{A}_L)$.

2. A follower's type $\theta_t \in \Theta$ is sampled according to $\mu$, i.e., $\theta_t \sim \mu$, and the follower plays action $a_{\theta_t}^\star(x_t)$.

3. Under *type feedback*, the leader observes both $a_{\theta_t}^\star(x_t)$ and $\theta_t$, while under *action feedback*, the leader only observes the best response $a_{\theta_t}^\star(x_t)$.

4. The leader samples and plays an action $a_i \sim x_t$. Then, the leader collects a utility of $u^L(a_i, a_{\theta_t}^\star(x_t))$, while the follower gets a utility of $u_{\theta_t}^F(a_i, a_{\theta_t}^\star(x_t))$.

The goal of the leader is to design learning algorithms that maximize their cumulative utility over the $T$ rounds. The performance of an algorithm is evaluated in terms of the *cumulative (Stackelberg) regret*, which is defined as:

$$R_T := T \cdot \text{OPT} - \mathbb{E}\left[\sum_{t \in [T]} u^L(x_t, a_{\theta_t}^\star(x_t))\right],$$

where the expectation is taken over the randomness arising from the sampling of follower's types and from the possible randomization in the leader's algorithm. Our goal is to design no-regret learning algorithms, which prescribe a sequence of commitments $\{x_t\}_{t \in [T]}$ that results in the regret $R_T$ growing sublinearly in $T$, namely $R_T = o(T)$.

## 3. A Negative Result With Action Feedback

We start with a negative result for the *action-feedback* setting. Specifically, we show that no algorithm can achieve regret polynomial in the bit-complexity of the follower's payoffs. More formally, we establish the following result:

**Theorem 3.1.** *Let $L \in \mathbb{N}$ be the bit-complexity of the follower's payoffs. Under action feedback, for any learning algorithm, there exists a BSG instance with constant-sized leader/follower action sets and set of follower types, and $T = \Theta(2^{2L})$ rounds, such that $R_T \geq \Omega(2^{2L})$.*

Intuitively, in order to prove the lower bound we consider $\Theta(2^{2L})$ instances. Each instance is characterized by a subregion in which all the follower's types play the unique action that provides positive utility to the leader. These regions are designed to be disjoint across instances, and the leader receives identical feedback whenever it selects a commitment outside them. As a result, in the worst case, the leader is required to enumerate an exponential number of regions before identifying the one that yields positive utility.

Employing Yao's minimax principle, this translates into an exponential lower bound on the regret.

Theorem 3.1 has strong practical implications. Indeed, even under commonly used 32-bit integer representations, *i.e.*, when $L = 32$, the lower bound on the regret suffered by any algorithm is in the order of $2^{64}$, which is prohibitively large. This highlights that action feedback is insufficient to achieve meaningful regret guarantees.

## 4. No-Regret With Type Feedback

In this section, we present a no-regret algorithm that operates in the *type-feedback* setting. The pseudocode of our algorithm is provided in Algorithm 1. At a high level, the algorithm splits the $T$ rounds into epochs indexed by $h \in \mathbb{N}$. In each epoch $h$, it executes three different procedures, namely `Find-Types` (Algorithm 2), `Find-Partition` (Algorithm 3), and `Prune` (Algorithm 4).[5]

At Line 1, Algorithm 1 initializes the leader's decision space $\mathcal{X}_h$—comprising the commitments available to the leader during epoch $h$—by setting $\mathcal{X}_1 = \Delta(\mathcal{A}_L)$. Moreover, Algorithm 1 initializes the parameter $\epsilon_h \in (0, 1)$ by setting $\epsilon_1 = 1/2K$. As further discussed in the following, this parameter plays a crucial role in balancing the length of the different epochs against the optimality of the commitments chosen by the leader. Then, at Line 3, Algorithm 1 iterates over the different epochs $h \in \mathbb{N}$ and, at each epoch $h$, performs the three main operations described in the following.

At Line 4, Algorithm 1 invokes the `Find-Types` procedure (see Algorithm 2). This procedure takes as inputs the set $\mathcal{X}_h$ and the parameter $\epsilon_h$, and requires $\mathcal{O}(1/\epsilon_h^2)$ leader-follower interactions. The goal of this procedure is to build an empirical estimator $\widehat{\mu}_h$ of the distribution $\mu$ such that $\|\widehat{\mu}_h - \mu\|_\infty \leq \epsilon_h$. Furthermore, this procedure also aims at identifying a subset of types $\overline{\Theta}_h \subseteq \Theta$ with the guarantee that, if $\theta \in \overline{\Theta}_h$, then $\mu_\theta \geq \epsilon_h$. Subsequently, at Line 5, Algorithm 1 computes the set $\widetilde{\Theta}_h$ by taking the union of $\overline{\Theta}_h$ and $\widetilde{\Theta}_{h-1}$ (where we initially set $\widetilde{\Theta}_0 := \varnothing$). Intuitively, the set $\widetilde{\Theta}_h$ consists of those follower types whose best-response regions will be known by the end of epoch $h$.

As a second main step, at Line 6, Algorithm 1 invokes the `Find-Partition` procedure (see Algorithm 3) to learn the polytopes $\mathcal{P}(\boldsymbol{a}) \cap \mathcal{X}_h$ associated with each action profile $\boldsymbol{a} \in \mathcal{A}_F(\widetilde{\Theta}_h)$. Algorithm 3 employs an adaptation of a procedure developed by Bacchiocchi et al. (2025) for non-*Bayesian* Stackelberg games to learn a mapping $\mathcal{Y}_h$ from action profiles $\boldsymbol{a} \in \mathcal{A}_F(\widetilde{\Theta}_h)$ to the corresponding sets $\mathcal{P}(\boldsymbol{a}) \cap \mathcal{X}_h$. Formally, $\mathcal{Y}_h : \boldsymbol{a} \mapsto \mathcal{P}(\boldsymbol{a}) \cap \mathcal{X}_h$.

---

[5]In Algorithm 1 and in all its sub-procedures, we do not keep track of the current round $t \in [T]$. We assume that, whenever $t > T$, the algorithm stops its execution.

---

**Algorithm 1** No-Regret-Bayesian-Stackelberg

---

**Require:** $T \in \mathbb{N}, \delta \in (0,1)$

1: $\epsilon_1 \leftarrow 1/2K$, $\mathcal{X}_1 \leftarrow \Delta(\mathcal{A}_{\mathrm{L}})$, $\widetilde{\Theta}_0 \leftarrow \varnothing$
2: $\delta_1 \leftarrow \frac{\delta}{2\lceil \log_4(5T) \rceil}$, $\delta_2 \leftarrow \delta_1$
3: **for** $h = 1, 2, \ldots$ **do**
4:     $\overline{\Theta}_h, \widehat{\mu}_h \leftarrow$ Find-Types$(\mathcal{X}_h, \epsilon_h, \delta_1)$
5:     $\widetilde{\Theta}_h \leftarrow \widetilde{\Theta}_{h-1} \cup \overline{\Theta}_h$
6:     $\mathcal{Y}_h \leftarrow$ Find-Partition$(\mathcal{X}_h, \epsilon_h, \widetilde{\Theta}_h, \widetilde{\Theta}_{h-1}, \delta_2)$
7:     $\mathcal{X}_{h+1} \leftarrow$ Prune$(\mathcal{Y}_h, \epsilon_h, \widetilde{\Theta}_h, \widetilde{\Theta}_{h-1}, \widehat{\mu}_h)$
8:     $\epsilon_{h+1} \leftarrow \epsilon_h/2$
9: **end for**

---

At Line 7, Algorithm 1 invokes the Prune procedure (see Algorithm 4). The goal of this procedure is to exploit the estimator $\widehat{\mu}_h$ and the mapping $\mathcal{Y}_h$ to refine the leader's decision space, thereby obtaining a new decision space $\mathcal{X}_{h+1}$ for the subsequent epoch $h+1$ and ensuring that all leader commitments selected at that epoch are at most $\mathcal{O}(\epsilon_h)$-suboptimal. Finally, at Line 8, Algorithm 1 halves the parameter $\epsilon_h$, setting $\epsilon_{h+1}$ for the subsequent epoch.

**Structure of the Leader's Decision Space** It is crucial to notice that the leader's decision space $\mathcal{X}_h$ is computed as the union, over action profiles $\boldsymbol{a}_{h-1} \in \mathcal{A}_{\mathrm{F}}(\widetilde{\Theta}_{h-1})$, of polytopes $\mathcal{X}_h(\boldsymbol{a}_{h-1}) \subseteq \mathcal{P}(\boldsymbol{a}_{h-1})$ with pairwise zero-volume intersections. Formally, we have:

$$\mathcal{X}_h \coloneqq \bigcup_{\boldsymbol{a}_{h-1} \in \mathcal{A}_{\mathrm{F}}(\widetilde{\Theta}_{h-1})} \mathcal{X}_h(\boldsymbol{a}_{h-1}). \tag{1}$$

As further discussed in Section 4.3, for each $\boldsymbol{a}_{h-1}$, the set $\mathcal{X}_h(\boldsymbol{a}_{h-1})$ is the subset of $\mathcal{P}(\boldsymbol{a}_{h-1})$ in which the leader's expected utility is at most $\mathcal{O}(\epsilon_{h-1})$-suboptimal.[6] Furthermore, in order to guarantee that Equation (1) is well defined for every epoch $h \geq 1$, including the first one, we let $\widetilde{\Theta}_0 \coloneqq \varnothing$, $\mathcal{A}_{\mathrm{F}}(\widetilde{\Theta}_0) \coloneqq \{\perp\}$, and $\mathcal{X}_1(\perp) \coloneqq \Delta(\mathcal{A}_{\mathrm{L}})$.

### 4.1. The Find-Types Procedure

We now discuss the first procedure executed by Algorithm 1 at each epoch $h$, namely Find-Types, whose pseudocode is presented in Algorithm 2. The goal of this procedure is to compute an estimator $\widehat{\mu}_h$ of the prior distribution $\mu$ such that $\|\widehat{\mu}_h - \mu\|_\infty \leq \epsilon_h$, together with a subset of types $\overline{\Theta}_h \subseteq \Theta$ satisfying $\mu_\theta \geq \epsilon_h$ for all $\theta \in \overline{\Theta}_h$.

In order to achieve its goal, Algorithm 2 selects an arbitrary strategy in the leader's decision space $\mathcal{X}_h$ and commits to it for $T_{h,1} = O(1/\epsilon_h^2)$ rounds. By using the realized types observed during these $T_{h,1}$ rounds of leader-follower inter-

---

**Algorithm 2** Find-Types

---

**Require:** $\epsilon_h > 0, \varnothing \neq \mathcal{X}_h \subseteq \Delta(\mathcal{A}_{\mathrm{L}}), \delta_1 \in (0,1)$

1: $T_{h,1} \leftarrow \left\lceil \frac{1}{2\epsilon_h^2} \log\left(\frac{2K}{\delta_1}\right) \right\rceil$
2: Play any $x \in \mathcal{X}_h$ for $T_{h,1}$ rounds and observe $\theta_t \sim \mu$
3: Compute $\widehat{\mu}_h$ with the observed feedback
4: $\overline{\Theta} \leftarrow \{\theta \in \Theta \mid \widehat{\mu}_{h,\theta} \geq 2\epsilon_h\}$
5: **Return** $\widehat{\mu}_h, \overline{\Theta}$

---

actions, the algorithm constructs the estimator $\widehat{\mu}_h$. Based on this estimator, it then defines $\overline{\Theta}_h$ as the set of all follower types $\theta$ such that $\widehat{\mu}_{h,\theta} \geq 2\epsilon_h$.

Then, by a standard concentration argument, Algorithm 2 ensures that the following holds.

**Lemma 4.1.** *Let $\epsilon_h, \delta_1 \in (0,1)$ and let $\mathcal{X}_h \subseteq \Delta(\mathcal{A}_{\mathrm{L}})$ be non-empty. Then, with probability at least $1 - \delta_1$, Algorithm 2 computes:*

*(i) an estimator $\widehat{\mu}_h \in \Delta(\Theta)$ such that $\|\mu - \widehat{\mu}_h\|_\infty \leq \epsilon_h$;*
*(ii) a set of types $\overline{\Theta}_h \subseteq \Theta$ such that $\mu_\theta \geq \epsilon_h$ for every $\theta \in \overline{\Theta}_h$ and $\mu_\theta \leq 3\epsilon_h$ for every other type;*

*by using $\mathcal{O}\left(1/\epsilon_h^2 \log(K/\delta_1)\right)$ rounds.*

### 4.2. The Find-Partition Procedure

Now, we introduce and describe the Find-Partition procedure, whose pseudocode is reported in Algorithm 3. The goal of this procedure is to identify the mapping

$$\mathcal{Y}(\boldsymbol{a}_h) \coloneqq \mathcal{P}(\boldsymbol{a}_h) \cap \mathcal{X}_h,$$

for every action profile $\boldsymbol{a}_h \in \mathcal{A}_{\mathrm{F}}(\widetilde{\Theta}_h)$. Equivalently, since the set $\mathcal{X}_h$ is defined as the union of polytopes $\mathcal{X}_h(\boldsymbol{a}_{h-1}) \subseteq \mathcal{P}(\boldsymbol{a}_{h-1})$ over the action profiles $\boldsymbol{a}_{h-1} \in \mathcal{A}_{\mathrm{F}}(\Theta_{h-1})$, Algorithm 3 determines, for each action profile $\boldsymbol{a}_h \in \mathcal{A}_{\mathrm{F}}(\widetilde{\Theta}_h)$ and $\boldsymbol{a}_{h-1} = \boldsymbol{a}_h|\widetilde{\Theta}_{h-1}$, the polytope:

$$\mathcal{Y}_h(\boldsymbol{a}_h) = \mathcal{P}(\boldsymbol{a}_h) \cap \mathcal{X}_h(\boldsymbol{a}_{h-1}). \tag{2}$$

More formally, given $\boldsymbol{a}_h$ and $\boldsymbol{a}_{h-1} = \boldsymbol{a}_h|\widetilde{\Theta}_{h-1}$, we have

$$\begin{aligned}
\mathcal{Y}_h(\boldsymbol{a}_h) &= \mathcal{P}(\boldsymbol{a}_h) \cap \mathcal{X}_h \\
&= \mathcal{P}(\boldsymbol{a}_h) \cap \bigcup_{\boldsymbol{a}'_{h-1} \in \mathcal{A}_{\mathrm{F}}(\Theta_{h-1})} \mathcal{X}_h(\boldsymbol{a}'_{h-1}) \\
&= \bigcup_{\boldsymbol{a}'_{h-1} \in \mathcal{A}_{\mathrm{F}}(\Theta_{h-1})} \left(\mathcal{P}(\boldsymbol{a}_h) \cap \mathcal{X}_h(\boldsymbol{a}'_{h-1})\right) \\
&= \mathcal{P}(\boldsymbol{a}_h) \cap \mathcal{X}_h(\boldsymbol{a}_{h-1}),
\end{aligned}$$

where the last equality holds since the intersection between $\mathcal{P}(\boldsymbol{a}_h)$ and $\mathcal{X}_h(\boldsymbol{a}'_{h-1})$ is empty for all action profiles $\boldsymbol{a}'_{h-1}$ that do not coincide with $\boldsymbol{a}_{h-1}$.[7]

---

[6]With a slight abuse of notation, we denote by $\boldsymbol{a}_h$ an element of $\mathcal{A}_{\mathrm{F}}(\widetilde{\Theta}_h)$ so as to distinguish it from action profiles associated with other epochs.

[7]For ease of exposition, in the following, whenever a polytope has zero volume, we consider it empty. This can be safely done as discussed in Appendix C.

In order to compute the mapping $\mathcal{Y}_h$, Algorithm 3 receives as inputs the parameter $\epsilon_h \in (0,1)$ such that $\mu_\theta \geq \epsilon_h$ for every $\theta \in \widetilde{\Theta}_h$ (according to Lemma 4.1), and the leader's decision space $\mathcal{X}_h$. In addition, Algorithm 3 takes as input a parameter $\delta_2 \in (0,1)$ that controls the probability with which it terminates correctly.

Algorithm 3 builds on a procedure developed by Bacchiocchi et al. (2025), which is designed to learn the best-response regions $\mathcal{P}_\theta(a)$ in the single-typed follower setting, *i.e.*, when $\Theta$ is a singleton. This procedure requires access to an oracle that maps each leader's commitment $x \in \Delta(\mathcal{A}_\mathrm{L})$ to the corresponding follower best response $a_\theta^\star(x)$. In our setting, such an oracle can be implemented by repeatedly letting the leader commit to the same mixed strategy $x$ until a follower of type $\theta$ is sampled. In such a case, we say that the oracle *queries* the strategy $x$. Notice that, since $\mu_\theta \geq \epsilon_h$ for every $\theta \in \widetilde{\Theta}_h$, with $\mathcal{O}(1/\epsilon_h)$ leader-follower interactions, the best response $a_\theta^\star(x)$ is observed at least once with high probability, allowing the oracle to effectively query the strategy $x$.

Let us also observe that the procedure in (Bacchiocchi et al., 2025) was originally designed to operate over the simplex $\Delta(\mathcal{A}_\mathrm{L})$. However, it can be easily adapted to work over a generic polytope $\mathcal{X}_h(\boldsymbol{a}_{h-1})$ (see, *e.g.*, (Bacchiocchi et al., 2024a)). Thus, the following holds.

**Lemma 4.2.** *[Informal restate (Bacchiocchi et al., 2025)] Let $\mathcal{X}_h(\boldsymbol{a}_{h-1})$ be a polytope with $N$ facets. Then, given any $\zeta \in (0,1)$ and $\theta \in \Theta$, under the event that each query terminates in a finite number of leader-follower interactions, there exists an algorithm that computes the polytopes $\mathcal{P}_\theta(a) \cap \mathcal{X}_h(\boldsymbol{a}_{h-1})$ for each $a \in \mathcal{A}_\mathrm{F}$ by using at most*

$$\widetilde{\mathcal{O}}\left(n^2\left(m^7 L \log\frac{1}{\zeta} + \binom{N+n}{m}\right)\right)$$

*queries with probability at least $1 - \zeta$.*

In the following, we denote by $\mathtt{Stackelberg}$ the procedure that ensures the validity of Lemma 4.2. This procedure takes as inputs a polytope $\mathcal{X}_h(\boldsymbol{a}_{h-1})$, a type $\theta \in \Theta$, and a parameter $\zeta \in (0,1)$, and returns the polytopes $\mathcal{P}_\theta(a \mid \mathcal{X}_h(\boldsymbol{a}_{h-1}))$ for $a \in \mathcal{A}_\mathrm{F}$, defined as follows:

$$\mathcal{P}_\theta(a \mid \mathcal{X}_h(\boldsymbol{a}_{h-1})) \coloneqq \mathcal{P}_\theta(a) \cap \mathcal{X}_h(\boldsymbol{a}_{h-1}), \quad (3)$$

with probability at least $1 - \zeta$. When $\mathcal{X}_h(\boldsymbol{a}_{h-1})$ is empty, the procedure simply returns an empty set, without requiring any leader-follower interactions.

Next, we describe Algorithm 3. At a high level, it consists of three macro blocks.

1. In the *first* macro block (Lines 3–7), Algorithm 3 computes the regions $\mathcal{P}_\theta(\,\cdot\mid \mathcal{X}_h(\boldsymbol{a}_{h-1}))$ for each type in $\widetilde{\Theta}_{h-1}$ and each action profile in $\mathcal{A}_\mathrm{F}(\widetilde{\Theta}_{h-1})$. We observe that, during the execution of Algorithm 1, given

---

**Algorithm 3** $\mathtt{Find\text{-}Partition}$

1: **Require** $\mathcal{X}_h, \widetilde{\Theta}_{h-1} \subseteq \widetilde{\Theta}_h \subseteq \Theta, \epsilon_h, \delta_2 \in (0,1)$
2: $\zeta \leftarrow \delta_2/2$
3: **for** $\theta \in \widetilde{\Theta}_{h-1}, \boldsymbol{a}_{h-1} \in \mathcal{A}_\mathrm{F}(\widetilde{\Theta}_{h-1})$ **do**
4: $\quad a \leftarrow \boldsymbol{a}_{h-1,\theta}$
5: $\quad \mathcal{P}_\theta(a \mid \mathcal{X}_h(\boldsymbol{a}_{h-1})) \leftarrow \mathcal{X}_h(\boldsymbol{a}_{h-1})$
6: $\quad \mathcal{P}_\theta(a' \mid \mathcal{X}_h(\boldsymbol{a}_{h-1})) \leftarrow \varnothing \quad \forall a' \neq a$
7: **end for**
8: **for** $\theta \in \widetilde{\Theta}_h \setminus \widetilde{\Theta}_{h-1}, \boldsymbol{a}_{h-1} \in \mathcal{A}_\mathrm{F}(\widetilde{\Theta}_{h-1})$ **do**
9: $\quad R \leftarrow \mathtt{Stackelberg}(\mathcal{X}_h(\boldsymbol{a}_{h-1}), \theta, \zeta)$
10: $\quad \mathcal{P}_\theta(a \mid \mathcal{X}_h(\boldsymbol{a}_{h-1})) \leftarrow R(a) \quad \forall a \in \mathcal{A}_\mathrm{F}$
11: **end for**
12: **for** $\boldsymbol{a}_h \in \mathcal{A}_\mathrm{F}(\widetilde{\Theta}_h)$ **do**
13: $\quad \boldsymbol{a}_{h-1} \leftarrow \boldsymbol{a}_h|_{\widetilde{\Theta}_{h-1}}$
14: $\quad$ **if** $\mathrm{vol}\left(\bigcap_{\theta \in \widetilde{\Theta}_h} \mathcal{P}_\theta(\boldsymbol{a}_{h,\theta} \mid \mathcal{X}_h(\boldsymbol{a}_{h-1}))\right) > 0$ **then**
15: $\quad\quad \mathcal{Y}_h(\boldsymbol{a}_h) \leftarrow \bigcap_{\theta \in \widetilde{\Theta}_h} \mathcal{P}_\theta(\boldsymbol{a}_{h,\theta} \mid \mathcal{X}_h(\boldsymbol{a}_{h-1}))$
16: $\quad$ **else**
17: $\quad\quad \mathcal{Y}_h(\boldsymbol{a}_h) \leftarrow \emptyset$
18: $\quad$ **end if**
19: **end for**
20: **Return** $\mathcal{Y}_h$

---

a tuple $\boldsymbol{a}_{h-1} \in \mathcal{A}_\mathrm{F}(\widetilde{\Theta}_{h-1})$, the following holds:

$$\mathcal{X}_h(\boldsymbol{a}_{h-1}) \subseteq \mathcal{P}(\boldsymbol{a}_{h-1}) \subseteq \mathcal{P}_\theta(\boldsymbol{a}_{h-1,\theta}). \quad (4)$$

The first inclusion follows from the definition of the leader decision space $\mathcal{X}_h$, while the second one follows from the definition of $\mathcal{P}(\boldsymbol{a}_{h-1})$. Taking $a = \boldsymbol{a}_{h-1,\theta}$ in Equation (3), thanks to Equation (4), we get:

$$\mathcal{P}_\theta(a \mid \mathcal{X}_h(\boldsymbol{a}_{h-1})) = \mathcal{X}_h(\boldsymbol{a}_{h-1}),$$

as implemented by Algorithm 3 at Line 5. Furthermore, for any action $a' \neq \boldsymbol{a}_{h-1,\theta}$, the intersection between $\mathcal{P}_\theta(\boldsymbol{a}_{h-1,\theta})$ and $\mathcal{P}_\theta(a')$ is either empty or has zero volume. Since $\mathcal{X}_h(\boldsymbol{a}_{h-1}) \subseteq \mathcal{P}_\theta(\boldsymbol{a}_{h-1,\theta})$ according to Equation (4), the region $\mathcal{P}_\theta(a' \mid \mathcal{X}_h(\boldsymbol{a}_{h-1}))$ is also either empty or has zero volume. Thus, it is set equal to $\varnothing$ at Line 6. We notice that Algorithm 3 does not require any leader-follower interactions here, as it simply manipulates the follower's best-response regions associated with follower types in $\widetilde{\Theta}_{h-1}$, which have been computed in previous epochs.

2. In the *second* macro block (Lines 8–11), Algorithm 3 invokes the $\mathtt{Stackelberg}$ procedure for each type in $\widetilde{\Theta}_h \setminus \widetilde{\Theta}_{h-1}$ and each action profile in $\mathcal{A}_\mathrm{F}(\widetilde{\Theta}_{h-1})$. By Lemma 4.2, the $\mathtt{Stackelberg}$ procedure is guaranteed to learn the polytope

$$R(a) \coloneqq \mathcal{P}_\theta(a) \cap \mathcal{X}_h(\boldsymbol{a}_{h-1}),$$

for every $a \in \mathcal{A}_\mathrm{F}$. According to Equation (3), the polytope $R(a)$ exactly coincides with $\mathcal{P}_\theta(a \mid \mathcal{X}_h(\boldsymbol{a}_{h-1}))$.

Thus, at Line 10, Algorithm 3 sets $\mathcal{P}_\theta(a \mid \mathcal{X}_h(\boldsymbol{a}_{h-1}))$ equal to $R(a)$, for every $a \in \mathcal{A}_\mathrm{F}$.

3. Finally, in the *third* macro block, Algorithm 3 computes $\mathcal{Y}_h(\boldsymbol{a}_h)$ for every $\boldsymbol{a}_h$ belonging to $\mathcal{A}_\mathrm{F}(\widetilde{\Theta}_h)$. We show that, for any $\boldsymbol{a}_h$ and $\boldsymbol{a}_{h-1}$ defined according to Line 13, the set $\mathcal{Y}_h(\boldsymbol{a}_h)$ can be computed as done at Line 15. Indeed, we have:

$$\begin{aligned}
\mathcal{Y}_h(\boldsymbol{a}_h) &= \mathcal{P}(\boldsymbol{a}_h) \cap \mathcal{X}_h(\boldsymbol{a}_{h-1}) \\
&= \Big( \bigcap_{\theta \in \widetilde{\Theta}_h} \mathcal{P}(\boldsymbol{a}_{h,\theta}) \Big) \cap \mathcal{X}_h(\boldsymbol{a}_{h-1}) \\
&= \bigcap_{\theta \in \widetilde{\Theta}_h} \Big( \mathcal{P}(\boldsymbol{a}_{h,\theta}) \cap \mathcal{X}_h(\boldsymbol{a}_{h-1}) \Big) \\
&= \bigcap_{\theta \in \widetilde{\Theta}_h} \mathcal{P}_\theta \big( \boldsymbol{a}_{h,\theta} \mid \mathcal{X}_h(\boldsymbol{a}_{h-1}) \big).
\end{aligned}$$

The first equality above holds because of the definition of Equation (2), the second one because of the definition of $\mathcal{P}(\boldsymbol{a}_h)$, the third one because of the distributive property of set intersection, and the last equality holds because of Equation (3).

As shown above, Algorithm 3 computes the mapping $\mathcal{Y}_h$ so long as the `Stackelberg` procedure correctly terminates every time it is invoked. By taking into account the number of rounds required to successfully execute the `Stackelberg` procedure with high probability (Lemma 4.2), we can provide the following guarantees.

**Lemma 4.3.** *[Informal version of Lemma C.3] Suppose that each polytope $\mathcal{X}_h(\boldsymbol{a})$ composing $\mathcal{X}_h$ has at most $N > 0$ facets. If $\mu_\theta \geq \epsilon$ for every $\theta \in \widetilde{\Theta}_h$, Algorithm 3 computes $\mathcal{Y}_h$ according to Equation (2) with probability at least $1 - \delta_2$, using at most:*

$$\widetilde{\mathcal{O}}\left( \frac{1}{\epsilon_h} K^{m+1} n^{2m+2} \left( m^7 L \log^2\left(\frac{1}{\delta_2}\right) + \binom{N+n}{m} \right) \right)$$

*rounds.*

We observe that, in principle, the number of action profiles $\boldsymbol{a}_{h-1} \in \mathcal{A}_\mathrm{F}(\widetilde{\Theta}_{h-1})$ is exponential in $K$, and Algorithm 3 would therefore require a number of rounds exponential in $K$. However, this is *not* the case, and Algorithm 3 exhibits an exponential dependence only on $m$. Intuitively, this is because most regions $\mathcal{X}_h(\boldsymbol{a}_{h-1})$ are empty, and thus executing the `Stackelberg` procedure over them requires no leader-follower interactions. More formally, it is possible to show that the number of non-empty best-response regions $\mathcal{P}(\boldsymbol{a}_{h-1})$ is bounded by $K^m n^{2m}$ (see (Personnat et al., 2025, Lemma 3.2)). Thus, since $\mathcal{X}_h(\boldsymbol{a}_{h-1}) \subseteq \mathcal{P}(\boldsymbol{a}_{h-1})$, we can show that the `Stackelberg` procedure is actually executed at most $\mathcal{O}(K^m n^{2m})$ times.

### 4.3. The `Prune` Procedure

We now discuss the last subprocedure executed at each epoch $h$ by Algorithm 1, whose pseudocode is reported in Algorithm 4. The goal of this procedure is to compute a decision space $\mathcal{X}_{h+1}$ for the subsequent epoch such that, for every $x \in \mathcal{X}_{h+1}$, it holds $u^\mathrm{L}(x) \geq \mathrm{OPT} - \mathcal{O}(K\epsilon_h)$. To do so, Algorithm 4 takes as inputs the mapping $\mathcal{Y}_h$ computed by Algorithm 3, the current set of types $\widetilde{\Theta}_h$, the estimator $\widehat{\mu}_h$ computed by Algorithm 2, and the precision level $\epsilon_h$.

During its execution, Algorithm 4 relies on empirical estimates of the leader's expected utility. Specifically, for every $\boldsymbol{a}_h \in \mathcal{A}_\mathrm{F}(\widetilde{\Theta}_h)$ and $x \in \mathcal{Y}_h(\boldsymbol{a}_h)$, we define

$$\widehat{u}_h^\mathrm{L}(x, \boldsymbol{a}_h) \coloneqq \sum_{\theta \in \widetilde{\Theta}_h} \widehat{\mu}_{h,\theta} u^\mathrm{L}(x, \boldsymbol{a}_{h,\theta}). \tag{5}$$

We notice that these estimates ignore types $\theta \notin \widetilde{\Theta}_h$ and rely on the empirical estimator $\widehat{\mu}_h$, incurring an approximation error of at most $\mathcal{O}(K\epsilon_h)$ thanks to Lemma 4.1.

---

**Algorithm 4** `Prune`

---

**Require:** $\mathcal{Y}_h, \widetilde{\Theta}_h, \epsilon_h, \widehat{\mu}_h$.
1: $C_1 \leftarrow 3, C_2 \leftarrow 6$
2: $\underline{\mathrm{OPT}}_h \leftarrow$ Equation (6)
3: **for** $\boldsymbol{a} \in \mathcal{A}_\mathrm{F}(\widetilde{\Theta}_h)$ **do**
4:     $\mathcal{H}_h(\boldsymbol{a}) \leftarrow$ Equation (7)
5:     $\mathcal{X}_{h+1}(\boldsymbol{a}) \leftarrow \mathcal{Y}_h(\boldsymbol{a}) \cap \mathcal{H}_h(\boldsymbol{a})$
6:     **if** $\mathrm{vol}(\mathcal{X}_{h+1}(\boldsymbol{a})) = 0$ **then**
7:         $\mathcal{X}_{h+1}(\boldsymbol{a}) \leftarrow \varnothing$
8:     **end if**
9: **end for**
10: **Return** $\mathcal{X}_{h+1} = \bigcup_{\boldsymbol{a} \in \mathcal{A}_\mathrm{F}(\widetilde{\Theta}_h)} \mathcal{X}_{h+1}(\boldsymbol{a})$

---

We now describe how Algorithm 4 works. As a first step, it computes a lower bound $\underline{\mathrm{OPT}}_h$ on the value of an optimal commitment, defined as:

$$\underline{\mathrm{OPT}}_h \coloneqq \max_{\substack{\boldsymbol{a} \in \mathcal{A}_\mathrm{F}(\widetilde{\Theta}_h) \\ x \in \mathcal{Y}_h(\boldsymbol{a})}} \widehat{u}_h^\mathrm{L}(x, \boldsymbol{a}) - C_2 K \epsilon_h, \tag{6}$$

where $C_2$ is a constant defined at Line 1. The procedure then computes the polytope $\mathcal{X}_{h+1}(\boldsymbol{a}_h)$, for every $\boldsymbol{a}_h \in \mathcal{A}_\mathrm{F}(\widetilde{\Theta}_h)$, by *pruning* the subset of $\mathcal{Y}_h(\boldsymbol{a}_h)$ that is guaranteed to yield low utility. To this end, it defines the half-spaces $\mathcal{H}_h(\boldsymbol{a}_h) \subseteq \mathbb{R}^m$ as follows:

$$\mathcal{H}_h(\boldsymbol{a}_h) \coloneqq \{x \in \mathbb{R}^m \mid \widehat{u}_h^\mathrm{L}(x, \boldsymbol{a}_h) + C_1 K \epsilon_h \geq \underline{\mathrm{OPT}}_h\}, \tag{7}$$

where $C_1$ is a constant defined at Line 1. Then, for every $\boldsymbol{a}_h \in \mathcal{A}_\mathrm{F}(\widetilde{\Theta}_h)$, the polytope $\mathcal{X}_{h+1}(\boldsymbol{a}_h)$ is defined as the intersection of the half-space $\mathcal{H}_h(\boldsymbol{a}_h)$ and $\mathcal{Y}_h(\boldsymbol{a}_h)$, *i.e.*, $\mathcal{X}_{h+1}(\boldsymbol{a}_h) \coloneqq \mathcal{Y}_h(\boldsymbol{a}_h) \cap \mathcal{H}_h(\boldsymbol{a}_h)$, as implemented at Line 5.

The half-spaces $\mathcal{H}_h(\boldsymbol{a}_h)$ defined in Equation (7) guarantee two crucial properties: (i) an optimal commitment belongs to at least one such $\mathcal{H}_h(\boldsymbol{a}_h)$ and therefore to the new search space $\mathcal{X}_{h+1}$, and (ii) all commitments belonging to such half-spaces are $\mathcal{O}(K\epsilon_h)$-optimal. Formally, we have:

**Lemma 4.4.** *[Informal version of Lemma D.5] Suppose that Algorithm 2 and Algorithm 3 were executed successfully up to epoch h. Then Algorithm 4 computes a search space*

$$\mathcal{X}_{h+1} = \bigcup_{\boldsymbol{a}_h \in \mathcal{A}_{\mathrm{F}}(\widetilde{\Theta}_h)} \mathcal{X}_{h+1}(\boldsymbol{a}_h)$$

*containing an optimal commitment. Furthermore, it holds $u^{\mathrm{L}}(x) \geq \mathrm{OPT} - 14K\epsilon_h$ for every $x \in \mathcal{X}_{h+1}$.*

To prove the first part of the lemma, we define:

$$x^\circ, \boldsymbol{a}^\circ \in \underset{\substack{\boldsymbol{a} \in \mathcal{A}_{\mathrm{F}}(\widetilde{\Theta}_h) \\ x \in \mathcal{Y}_h(\boldsymbol{a})}}{\arg\max} \widehat{u}_h^{\mathrm{L}}(x, \boldsymbol{a}).$$

Then, we can prove that the following holds:

$$
\begin{aligned}
\widehat{u}_h^{\mathrm{L}}(x^\star, \boldsymbol{a}^\star) &\geq u^{\mathrm{L}}(x^\star) - C'K\epsilon_h \\
&\geq u^{\mathrm{L}}(x^\circ) - C'K\epsilon_h \\
&\geq \widehat{u}_h^{\mathrm{L}}(x^\circ, \boldsymbol{a}^\circ) - C''K\epsilon_h \\
&= \widehat{u}_h^{\mathrm{L}}(x^\circ, \boldsymbol{a}^\circ) \pm C_2 K\epsilon_h - C''K\epsilon_h \\
&= \underline{\mathrm{OPT}}_h - (C'' - C_2)K\epsilon_h.
\end{aligned}
$$

where $C', C'' > 0$ are two absolute constants defined in Appendix D. Notice that the first and the third inequality above hold because of Lemma 4.1, while the second inequality holds because of the optimality of $x^\star$. Finally, by choosing $C_1 \geq C'' - C_2$, we have that an optimal commitment always satisfies Equation (7) and thus belongs to $\mathcal{X}_{h+1}$.

The proof of the second part of the lemma, *i.e.*, the one showing that every commitment in $\mathcal{X}_{h+1}$ is $\mathcal{O}(K\epsilon_h)$-optimal, is provided in Appendix D and follows an argument similar to the one presented above.

### 4.4. Theoretical Guarantees

Before presenting the regret guarantees of Algorithm 1 we introduce two useful lemmas. The first lemma provides an upper bound on the number of facets defining the regions $\mathcal{X}_h(\boldsymbol{a}_{h-1})$ for each tuple $\boldsymbol{a}_{h-1} \in \widetilde{\Theta}_{h-1}$ and epoch $h$. Formally, we have:

**Lemma 4.5.** *[Informal version of Lemma E.6] During the execution of Algorithm 1, every polytope $\mathcal{X}_h(\boldsymbol{a}_{h-1})$ has at most $N \leq Kn + m + K$ facets as long as Algorithm 2 and Algorithm 3 are always executed successfully.*

Notice that, thanks to Lemma 4.5, the binomial term affecting the number of leader-follower interactions required to execute Algorithm 3 is polynomial once $m$ is fixed.

The second lemma provides an upper bound on the largest epoch index $H$ executed by Algorithm 1 over $T$ rounds.

**Lemma 4.6.** *The largest epoch index $H \in \mathbb{N}$ executed by Algorithm 1 satisfies $H \leq H' := \log_4(5T)$.*

To prove Lemma 4.6, we observe the following. Let $T_h$ be the number of rounds required to execute epoch $h$. Then, the following holds:

$$T = \sum_{h=1}^{H} T_h \geq \sum_{h=1}^{H-1} T_h \geq \sum_{h=1}^{H-1} \frac{1}{\epsilon_h^2}, \tag{8}$$

where the last inequality follows from the fact that $T_h$ is always larger than the number of rounds required to execute Algorithm 2 for each $h \in [H-1]$. Using the definition $\epsilon_h := 2^{-h}/K$ and the geometric series, rearranging Equation (8) yields the statement of Lemma 4.6. Then, we can prove that the following holds.

**Theorem 4.7.** *[Informal version of Theorem E.7] With probability at least $1 - \delta$, the regret of Algorithm 1 is:*

$$R_T \leq \widetilde{\mathcal{O}}\left(K^2 \log\left(\frac{1}{\delta}\right)\sqrt{T} + \beta\log^2(T)\right),$$

*where $\beta := \mathrm{poly}(n, K, L, \log(1/\delta))$ when $m$ is constant.*

We now provide a proof sketch of the above theorem. For the sake of presentation, in the following we omit the dependence on logarithmic terms and we assume $m$ to be a constant. A complete proof, including the omitted technical details, is deferred to the appendix. We observe that the number of rounds $T_h$ required to execute a generic epoch $h$ of Algorithm 1 can be bounded as follows:

$$T_h \leq \mathcal{O}\left(\underbrace{\frac{1}{\epsilon_h^2}}_{\text{Algorithm 2}} + \underbrace{\frac{1}{\epsilon_h} \cdot \mathrm{poly}(n, K, L)}_{\text{Algorithm 3}}\right), \tag{9}$$

according to Lemma 4.1 and Lemma 4.3, together with Lemma 4.5, and the fact that Algorithm 4 does not require any leader-follower interactions. Therefore, the regret suffered by Algorithm 1 can be upper bounded as follows:

$$
\begin{aligned}
R_T &\leq \mathcal{O}\left(\sum_{h=1}^{H'} T_h K\epsilon_{h-1}\right) \\
&\leq \mathcal{O}\left(\sum_{h=1}^{H'}\left(\frac{K}{\epsilon_h} + \mathrm{poly}(n, K, L)\right)\right) \\
&\leq \mathcal{O}\left(K^2\sqrt{T} + \mathrm{poly}(n, K, L)\log(T)\right).
\end{aligned}
$$

The first inequality follows from Lemma 4.4, which ensures that each commitment selected at epoch $h$ is $\mathcal{O}(K\epsilon_{h-1})$-optimal, and Lemma 4.6. The second inequality follows

by employing Equation (9) and observing that $\epsilon_{h-1} = 2\epsilon_h$. Finally, the third inequality follows by setting the precision parameter as $\epsilon_h = 2^{-h}/K$ and employing the geometric series together with Lemma 4.6.

**On The Tightness of Theorem 4.7** Thanks to Theorem 4.7, Algorithm 1 achieves a $\widetilde{\mathcal{O}}(\sqrt{T})$ regret upper bound when the number of the leader's actions $m$ is a fixed constant. Personnat et al. (2025) show that, even when the leader knows the payoff function of each type, our problem admits an $\Omega(\sqrt{T})$ lower bound on the regret achievable by any algorithm. Conversely, when $m$ is *not* fixed, Theorem 4.7 exhibits an exponential dependence on $m$ in the regret suffered by Algorithm 1. However, as shown by Peng et al. (2019), even in the case of a single follower's type, such an exponential dependence is unavoidable.

**On the Running Time of Algorithm 1** The algorithms presented in this paper can be implemented with per-round running time polynomial in $K$ and $n$ when $m$ is constant, an assumption that is already required to avoid exponential regret. An efficient implementation relies on two key observations: at every epoch $h$ there are at most $\mathcal{O}(K^m n^{2m})$ non-empty regions $\mathcal{X}_h(\boldsymbol{a}_h)$, and each one is determined by $O(Kn)$ hyperplanes. This observation allows to implement the algorithm in polynomial time and space complexity. One needs only to keep track of the non-empty regions, and to iterate over them rather than over every action profile when prescribed by the pseudocode. Furthermore, the exponential running time when $m$ is not constant is expected, as the offline problem of computing an optimal commitment in a BSG is NP-hard in such a case. Indeed, the main bottleneck of Algorithm 1 is the computation of $\underline{\mathrm{OPT}}_h$ in Algorithm Prune, which requires solving at most $\mathcal{O}(K^m n^{2m})$ LPs. This is the same computational cost required to solve the offline problem.

## Acknowledgements

This publication was funded with the contribution of Ministero dell'Università e della ricerca pursuant to D.D. n. 18010 of 12 November 2025 – BANDO FIS 3. Project FIS-2024-05736 (Starting Grant), title: "Towards a trustworthy strategic use of data in machine learning pipelines" (STRATDATA). CUP: D53C25002380001

## Impact Statement

This paper presents work whose goal is to advance the field of Machine Learning. There are many potential societal consequences of our work, none of which we feel must be specifically highlighted here.

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

# A. Proofs Omitted from Section 3

**Theorem 3.1.** *Let $L \in \mathbb{N}$ be the bit-complexity of the follower's payoffs. Under action feedback, for any learning algorithm, there exists a BSG instance with constant-sized leader/follower action sets and set of follower types, and $T = \Theta(2^{2L})$ rounds, such that $R_T \geq \Omega(2^{2L})$.*

*Proof.* **Building the instances** We consider a set of instances $\mathcal{I}$ in which the leader's action set is $\mathcal{A}_{\mathrm{L}} \coloneqq \{a_1, a_2, a_3\}$ and the follower's action set is $\mathcal{A}_{\mathrm{F}} \coloneqq \{a_1, a_2, a_3, a^\star\}$, and the leader's utility function is defined as follows.

$$u^{\mathrm{L}}(a_i, a_j) = \begin{cases} 1 & a_j = a^\star, \\ 0 & a_j \neq a^\star. \end{cases}$$

The set of types is $\Theta \coloneqq \{\theta_1, \theta_2, \theta_3\}$, with uniform prior distribution $\mu$. Each instance $\mathrm{I} \in \mathcal{I}$ is parametrized by a subsimplex $\mathcal{S}^{\mathrm{I}} \subseteq \Delta(\mathcal{A}_{\mathrm{F}})$, corresponding to the set of optimal leader strategies in that specific instance (see Figure 1). The subsimplex

$$\mathcal{S}^{\mathrm{I}} \coloneqq \Delta(\mathcal{A}_{\mathrm{L}}) \cap \{\mathcal{H}_j^{\mathrm{I}}\}_{j \in [m]}$$

is defined as the intersection of $\Delta(\mathcal{A}_{\mathrm{L}})$ with $m$ half-spaces $\mathcal{H}_j^{\mathrm{I}}$, for $j \in [m]$, defined as follows:

$$\mathcal{H}_j^{\mathrm{I}} = \{x \in \mathbb{R}^m \mid \sum_{i \in [m]} w_{j,i}^{\mathrm{I}} x_i \geq 0\}, \tag{10}$$

where $w_{j,i}^{\mathrm{I}} \in [-1/2, 1/2]$ has bit complexity bounded by $CB$, for some $B \in \mathbb{N}$ and a universal constant $C > 0$. We refer the reader to the next paragraph for the formal definition of these subsimplices.

Given an instance $\mathrm{I} \in \mathcal{I}$ and a corresponding subsimplex $\mathcal{S}^{\mathrm{I}}$, we show how to define the follower's utility function so that every commitment $x \in \mathcal{S}^{\mathrm{I}}$ is optimal, while all other commitments are suboptimal. To do so, we need to ensure that $\mathcal{P}_\theta(a^\star) = \mathcal{S}^{\mathrm{I}}$ for every type $\theta \in \Theta$. In this way, the leader achieves utility $u^{\mathrm{L}}(x) = 1$ when $x \in \mathcal{S}^{\mathrm{I}}$, and utility $u^{\mathrm{L}}(x) = 0$ when $x \notin \mathcal{S}^{\mathrm{I}}$.

We start by considering the first type $\theta_1 \in \Theta$, and devise the utility function $u_{\theta_1}^{\mathrm{F,I}} : \mathcal{A}_{\mathrm{L}} \times \mathcal{A}_{\mathrm{F}} \to [0, 1]$ in such a way that $\mathcal{P}_{\theta_1}(a^\star) = \mathcal{S}^{\mathrm{I}}$. In order to do so, the follower's utilities must satisfy the following:

$$\begin{cases} u_{\theta_1}^{\mathrm{F,I}}(a_i, a^\star) - u_{\theta_1}^{\mathrm{F,I}}(a_i, a_j) = w_{j,i}^{\mathrm{I}} & \forall a_j \in \mathcal{A}_{\mathrm{L}} \setminus \{a^\star\}, \\ 0 \leq u_{\theta_1}^{\mathrm{F,I}}(a_i, a_j) \leq 1 & \forall a_j \in \mathcal{A}_{\mathrm{F}}, a_i \in \mathcal{A}_{\mathrm{L}}. \end{cases} \tag{11}$$

Then, the follower's utility function for type $\theta_1 \in \Theta$ in instance $I \in \mathcal{I}$ is defined as follows:

$$\begin{cases} u_{\theta_1}^{\mathrm{F,I}}(a_i, a^\star) = \frac{1}{2} & \forall a_i \in \mathcal{A}_{\mathrm{L}} \\ u_{\theta_1}^{\mathrm{F,I}}(a_i, a_j) = \frac{1}{2} - w_{i,j}^{\mathrm{I}} & \forall a_i \in \mathcal{A}_{\mathrm{L}}, a_j \in \mathcal{A}_{\mathrm{F}} \setminus \{a^\star\}. \end{cases}$$

With a simple calculation, it is easy to verify that the above definition of the follower's utility satisfies Equation (11). We complete the instance by defining the utility functions for types $\{\theta_2, \theta_3\}$. In particular, for $k \in \{2, 3\}$ and $a_i \in \mathcal{A}_{\mathrm{L}}$, we let:

$$\begin{cases} u_{\theta_k}^{\mathrm{F,I}}(a_i, a^\star) = u_{\theta_1}^{\mathrm{F,I}}(a_i, a^\star), \\ u_{\theta_k}^{\mathrm{F,I}}(a_i, a_j) = u_{\theta_1}^{\mathrm{F,I}}(a_i, a_{f(j,k)}) & \forall a_j \in \mathcal{A}_{\mathrm{F}} \setminus \{a^\star\}, \end{cases}$$

where $f(j, k) \coloneqq 1 + ((j + k + 1) \mod 3)$.

It is easy to see that $\mathcal{P}_{\theta_k}(a^\star) = \mathcal{S}^{\mathrm{I}}$ for every $k \in [3]$. Therefore, $a_\theta^\star(x) = a^\star$ for every $x \in \mathcal{S}^{\mathrm{I}}$ and every $\theta \in \Theta$. Instead, for every $x \notin \mathcal{S}^{\mathrm{I}}$ we have that $a_\theta^\star(x) \neq a^\star$ for every $\theta \in \Theta$. Thus, if the leader plays $x \in \mathcal{S}^{\mathrm{I}}$, they observe action $a^\star$ with probability one, otherwise they observe an action $a$ drawn uniformly at random from $\{a_1, a_2, a_3\}$ (as $\mu$ is uniform over the types). Finally, we observe that the bit complexity of the follower's utility is bounded by $C'B$ for some constant $C' > C$.

**Building the optimal regions** Let $\epsilon = 2^{-B}$ for some $B > 0$ defined in the following and let $\mathcal{T}_\epsilon$ be the regular triangulation of the simplex into subsimplices of side $\epsilon > 0$ (see Figure 1). Each instance $\mathrm{I} \in \mathcal{I}$ is associated with a subsimplex $S^{\mathrm{I}} \in \mathcal{T}_\varepsilon$.

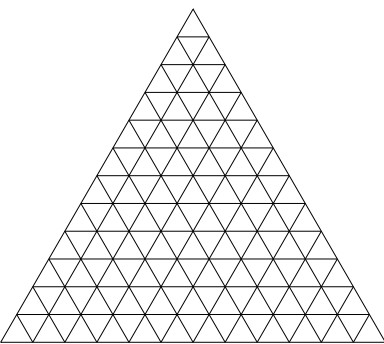

*Figure 1.* Each subsimplex represented in the figure is associated with a single instance of the lower bound.

Given an instance I, the subsimplex $S^{\mathrm{I}}$ has boundaries given by the hyperplanes defining the half-spaces $\{\mathcal{H}_j^{\mathrm{I}}\}_{j\in[m]}$, which we denote by $\{H_j^{\mathrm{I}}\}_{j\in[m]}$. Each $H_j^{\mathrm{I}}$ passes through the origin and two points on the boundary of the simplex, which for ease of exposition we take to be $(k\epsilon, 1 - k\epsilon, 0)$ and $(k\epsilon, 0, 1 - k\epsilon)$ for some $k \in \{0, 1, \ldots, 2^B\}$. The coefficients $w_j^{\mathrm{I}}$ of its algebraic representation satisfy the following:

$$k\epsilon w_{j,1}^{\mathrm{I}} + (1 - k\epsilon)w_{j,2}^{\mathrm{I}} + 0w_{j,3}^{\mathrm{I}} = 0$$
$$k\epsilon w_{j,1}^{\mathrm{I}} + 0w_{j,3}^{\mathrm{I}} + (1 - k\epsilon)w_{j,3}^{\mathrm{I}} = 0$$
$$-1/2 \le w_{j,i}^{\mathrm{I}} \le 1/2 \quad \forall i \in [m].$$

By an argument similar to the one provided in Lemma D.2 by (Bacchiocchi et al., 2025), the bit complexity of each $w_{j,i}^{\mathrm{I}}$ is at most $CB$ for some constant $C > 0$. Thus, the bit complexity of the follower's utility is bounded by $C'B$ for some $C' > C$.

**Lower bound on the regret** We observe that the number of instances satisfies $|\mathcal{T}_\epsilon| = \Theta(2^{2B})$. In each instance $\mathrm{I} \in \mathcal{I}$, the leader obtains utility equal to one and observes action $a^\star$ if they play $x \in S^{\mathrm{I}}$ (which is an optimal commitment). Otherwise, they collect zero utility and observe an action sampled uniformly at random from $\{a_1, a_2, a_3\}$. The behavior of an optimal deterministic algorithm is to play commitments belonging to the vertices of $\mathcal{T}_\epsilon$ according to some fixed order, and whenever it observes the optimal action $a^\star$, it starts playing it, since any commitment outside the optimal region does not help distinguish among remaining instances. The reason why an optimal algorithm should select vertices of the subsimplices is that, if it picks a vertex and does not observe action $a^\star$, it can exclude (at most) six instances as non-optimal, since the same vertex is a vertex of (at most) six subsimplices and the follower adopts optimistic tie-breaking.

Thus, given $T$ rounds, any deterministic algorithm can check at most $6T$ instances to determine whether they are optimal or not. Let $T = \frac{(1-3/4)}{6}|\mathcal{T}_\epsilon|$ be the number of rounds. Then, the probability that the algorithm never selects an optimal commitment over $T$ rounds is at least

$$1 - \frac{6T}{|\mathcal{T}_\epsilon|} = 1 - \frac{(1 - 3/4)|\mathcal{T}_\epsilon|}{|\mathcal{T}_\epsilon|} = \frac{3}{4}.$$

Therefore, the regret suffered by any deterministic algorithm is $\Theta(|\mathcal{T}_\epsilon|)$. By Yao's minimax principle, this implies that for any (possibly) randomized algorithm there exists an instance on which it suffers $\Theta(|\mathcal{T}_\epsilon|)$ regret. To conclude the proof, we set $B := \frac{L}{C'}$, so that $\Theta(|\mathcal{T}_\epsilon|) = \Theta(2^{2B}) = \Theta(2^{2L})$ and $T = \Theta(2^{2L})$. $\qquad\square$

## B. Omitted Proofs from Section 4.1

**Lemma 4.1.** *Let $\epsilon_h, \delta_1 \in (0, 1)$ and let $\mathcal{X}_h \subseteq \Delta(\mathcal{A}_{\mathrm{L}})$ be non-empty. Then, with probability at least $1 - \delta_1$, Algorithm 2 computes:*

*(i) an estimator $\widehat{\mu}_h \in \Delta(\Theta)$ such that $\|\mu - \widehat{\mu}_h\|_\infty \le \epsilon_h$;*
*(ii) a set of types $\overline{\Theta}_h \subseteq \Theta$ such that $\mu_\theta \ge \epsilon_h$ for every $\theta \in \overline{\Theta}_h$ and $\mu_\theta \le 3\epsilon_h$ for every other type;*

*by using $\mathcal{O}\left(1/\epsilon_h^2 \log(K/\delta_1)\right)$ rounds.*

*Proof.* Algorithm 2 computes the empirical estimator $\widehat{\mu}_h$ of $\mu$ using

$$T_1 := \left\lceil \frac{1}{2\epsilon_h^2} \log\left(\frac{2K}{\delta_1}\right) \right\rceil \tag{12}$$

samples. Subsequently it computes the set $\widetilde{\Theta} := \{\theta \in \Theta \mid \widehat{\mu}_{h,\theta} \geq 2\epsilon_h\}$.

Applying Hoeffding's inequality and a union bound we get that $\|\widehat{\mu}_{h,\theta} - \mu\|_\infty \leq \epsilon_h$ with probability at least $1 - \delta_1$. Therefore, with probability at least $1 - \delta_1$, when $\widehat{\mu}_{h,\theta} \geq 2\epsilon$ the true probability $\mu_\theta$ satisfies $\mu_\theta \geq \epsilon_h$, while when $\widehat{\mu}_{h,\theta} \leq 2\epsilon_h$, we have $\mu_\theta \leq 3\epsilon_h$. $\qquad\square$

# C. Details and Proofs Omitted from Section 4.2

## C.1. Details and Formal Theorems of Section 4.2

Lemma 4.2 and Lemma 4.3 in Section 4.2 omit some details for the sake of exposition. In this appendix we first describe these details and state the full formal lemmas, then we will provide their proofs. Specifically, these details concern (1) the exact shape of the polytopes $\mathcal{P}_\theta(a \mid \mathcal{X}_h(\boldsymbol{a}_{h-1}))$ and $\mathcal{Y}_h(\boldsymbol{a}_h)$ computed by Algorithm 3, (2) the exact dependence on the bit-complexity, and (3) the exact conditions that must be fulfilled to correctly execute the algorithm.

In Section 4.1, we defined the polytopes $\mathcal{P}_\theta(a \mid \mathcal{X}_h(\boldsymbol{a}_{h-1}))$ according to Equation (3) and $\mathcal{Y}_h(\boldsymbol{a}_h)$ according to Equation (2). However, Algorithm 3 does not compute these polytopes *exactly*. Instead, whenever Equation (3) or Equation (2) defines a non-empty polytope with null volume, Algorithm 3 considers it to be empty. Hence, it computes the empty set instead of a lower-dimensional polytope. In the following, with an abuse of notation we redefine these regions so that they cannot be lower dimensional polytopes. Specifically, given a polytope $\mathcal{S} \subseteq \Delta(\mathcal{A}_\mathrm{L})$, an action $a \in \mathcal{A}_\mathrm{F}$ and a type $\theta \in \Theta$, we let:

$$\mathcal{P}_\theta(a \mid S) := \begin{cases} \mathcal{P}_\theta(a) \cap S & \text{if } \mathrm{vol}(\mathcal{P}_\theta(a) \cap S) > 0 \\ \emptyset & \text{otherwise.} \end{cases} \tag{13}$$

The procedure by (Bacchiocchi et al., 2025) computes polytopes according to Equation (13) rather than Equation (3). Similarly, we define:

$$\mathcal{Y}_h(\boldsymbol{a}) := \begin{cases} \mathcal{P}(\boldsymbol{a}) \cap \mathcal{X}_h(\boldsymbol{a}_{h-1}) & \text{if } \mathrm{vol}(\mathcal{P}(\boldsymbol{a}) \cap \mathcal{X}_h(\boldsymbol{a}_{h-1})) > 0 \\ \emptyset & \text{otherwise,} \end{cases} \tag{14}$$

for every $\boldsymbol{a}_h \in \widetilde{\Theta}_h$, with $\boldsymbol{a}_{h-1} = \boldsymbol{a}_h|\widetilde{\Theta}_{h-1}$. Algorithm 3 computes $\mathcal{Y}_h$ according to the equation provided above rather than Equation 2.

The second detail that we need to clarify regards the relationship between the bit-complexity of the quantities managed by Algorithm 3 and the number of samples it requires. Specifically, the sample-complexity of Algorithm 3 is linear in the bit-complexity $L$ of the follower's payoffs. Moreover, it also depends on the number of bits required to encode the coefficients of the hyperplanes defining the region $\mathcal{X}_h(\boldsymbol{a}_{h-1})$, $\boldsymbol{a}_{h-1} \in \widetilde{\Theta}_h$. In the formal versions of Lemma 4.2 and Lemma 4.3, we will let $B > L$ be an upper bound on the number of bits required to encode each of these coefficients. According to this definition, the number of rounds required by Algorithm 3 scales linearly in $B$.

The last detail that we need is the condition that the parameters in input to Algorithm 3 must satisfy, which we formalize as follows.

**Condition C.1.** *Algorithm 3 is called with the following inputs: parameters $\epsilon_h, \delta_2 \in (0, 1)$, sets of types $\widetilde{\Theta}_{h-1} \subseteq \widetilde{\Theta}_h \subseteq \Theta$ such that $\mu_\theta \geq \epsilon_h$ for every $\theta \in \widetilde{\Theta}_h \setminus \widetilde{\Theta}_{h-1}$, search space*

$$\mathcal{X}_h = \bigcup_{\boldsymbol{a}_{h-1} \in \mathcal{A}_\mathrm{F}(\widetilde{\Theta}_{h-1})} \mathcal{X}_h(\boldsymbol{a}_{h-1}),$$

*where each $\mathcal{X}_h(\boldsymbol{a}_{h-1}) \subseteq \mathcal{P}(\boldsymbol{a}_{h-1})$ is empty or a non-zero-volume polytope.*

Intuitively, Condition C.1 requires that Algorithm 2 and Algorithm 3 have been correctly executed every previous epoch. In Appendix E we will show that this condition is always verified with high probability.

Finally, we can state the formal versions of Lemma 4.2 and Lemma 4.3. Specifically, the algorithm by (Bacchiocchi et al., 2025) provides the following guarantees.

**Lemma C.2.** *[Restate (Bacchiocchi et al., 2025), formal version of Lemma 4.2] Let $S \subseteq \Delta(\mathcal{A}_\mathrm{L})$ be a polytope with $\mathrm{vol}(S) > 0$ defined as the intersections of $N$ hyperplanes whose coefficients have bit complexity bounded by some $B \geq L$. Then, given any $\zeta \in (0,1)$ and $\theta \in \Theta$, under the event that each query terminates in a finite number rounds, there exists an algorithm that computes the polytopes $\mathcal{P}_\theta(a \mid S)$ according to Equation (13) for each $a \in \mathcal{A}_\mathrm{F}$ by using at most*

$$\widetilde{\mathcal{O}}\left(n^2\left(m^7 B \log\left(\frac{1}{\zeta}\right) + \binom{N+n}{m}\right)\right)$$

*queries with probability at least $1 - \zeta$.*

We will say that an execution of `Stackelberg` is *successful* when it computes the polytopes $\mathcal{P}_\theta(a|S)$ according to Equation (13) in the number of rounds specified in the lemma above. The formal version of Lemma 4.3 is the following.

**Lemma C.3.** *[Formal version of Lemma 4.3] Suppose that Condition C.1 is satisfied and each polytope $\mathcal{X}_h(\boldsymbol{a})$ composing $\mathcal{X}_h$ is defined as the intersections of $N$ hyperplanes whose coefficients have bit complexity bounded by some $B \geq L$. Then Algorithm 3 computes $\mathcal{Y}_h$ according to Equation (14) with probability at least $1 - \delta_2$, using at most:*

$$\widetilde{\mathcal{O}}\left(\frac{1}{\epsilon_h} K^{m+1} n^{2m+2}\left(m^7 B \log^2\left(\frac{1}{\delta_2}\right) + \binom{N+n}{m}\right)\right)$$

*rounds.*

### C.2. Intermediate Lemmas and Proof of Lemma 4.3

To prove Lemma C.3 (formal version of Lemma 4.3), we first provide two intermediate lemmas. These two results show that, when the `Stackelberg` subprocedure is always executed correctly, Algorithm 3 computes $\mathcal{Y}_h$ according to Equation (14). The proof of Lemma C.3 follows by bounding the number of samples required by such a procedure and the probability of correctly executing it.

**Lemma C.4.** *Suppose that Condition C.1 is satisfied and every execution of `Stackelberg` is successful, Algorithm 3 computes correctly $\mathcal{P}_\theta(a \mid \mathcal{X}_h(\boldsymbol{a}_{h-1}))$ according to Equation 13 for every $\theta \in \widetilde{\Theta}_h$, $a \in \mathcal{A}_\mathrm{F}$ and $\boldsymbol{a}_{h-1} \in \mathcal{A}_\mathrm{F}(\widetilde{\Theta}_{h-1})$.*

*Proof.* For the types $\theta \in \widetilde{\Theta}_h \setminus \widetilde{\Theta}_{h-1}$, the algorithm computes $\mathcal{P}_\theta(a \mid \mathcal{X}_h(\boldsymbol{a}_{h-1}))$ by means of the `Stackelberg` subprocedure, which computes it according to Equation (14) when successful. There remains to be considered the types in $\widetilde{\Theta}_{h-1}$. Let $\theta \in \widetilde{\Theta}_{h-1}$, $a \in \mathcal{A}_\mathrm{F}$ and $\boldsymbol{a}_{h-1} \in \mathcal{A}_\mathrm{F}(\widetilde{\Theta}_{h-1})$. In the following we let $\mathcal{P}_\theta(a \mid \mathcal{X}_h(\boldsymbol{a}_{h-1}))$ be defined according to Equation (13) and $\mathcal{P}'_\theta(a \mid \mathcal{X}_h(\boldsymbol{a}_{h-1}))$ be the region computed by Algorithm 3 at Lines 5 and 6.

Suppose that $a = \boldsymbol{a}_{h-1,\theta}$. Then Algorithm 3 computes $\mathcal{P}'_\theta(a \mid \mathcal{X}_h(\boldsymbol{a}_{h-1})) = \mathcal{X}_h(\boldsymbol{a}_{h-1})$. Condition C.1 guarantees that $\mathcal{X}_h(\boldsymbol{a}_{h-1}) \subseteq \mathcal{P}(\boldsymbol{a}_{h-1})$. As a result, we have that:

$$\mathcal{P}'_\theta(a \mid \mathcal{X}_h(\boldsymbol{a}_{h-1})) = \mathcal{X}_h(\boldsymbol{a}_{h-1}) = \mathcal{X}_h(\boldsymbol{a}_{h-1}) \cap \mathcal{P}(\boldsymbol{a}_{h-1})$$

Observe that such a polytope $\mathcal{X}_h(\boldsymbol{a}_{h-1}) = \mathcal{X}_h(\boldsymbol{a}_{h-1}) \cap \mathcal{P}(\boldsymbol{a}_{h-1})$ cannot have zero volume while being non-empty. Indeed, $\mathcal{X}_h(\boldsymbol{a}_{h-1})$ is either empty or has strictly positive volume, as required by Condition C.1. Therefore, $\mathcal{P}'_\theta(a \mid \mathcal{X}_h(\boldsymbol{a}_{h-1})) = \mathcal{P}_\theta(a \mid \mathcal{X}_h(\boldsymbol{a}_{h-1}))$ when $a = \boldsymbol{a}_{h-1,\theta}$.

Suppose instead that $a \neq \boldsymbol{a}_{h-1,\theta}$. By construction, Algorithm 3 computes $\mathcal{P}'_\theta(a|\mathcal{X}_h(\boldsymbol{a}_{h-1})) = \emptyset$. At the same time, we have that $\mathcal{X}_h(\boldsymbol{a}_{h-1}) \subseteq \mathcal{P}(\boldsymbol{a}_{h-1}) \subseteq \mathcal{P}_\theta(\boldsymbol{a}_{h-1,\theta})$ and $\mathrm{vol}(\mathcal{P}_\theta(\boldsymbol{a}_{h-1,\theta}) \cap \mathcal{P}_\theta(a)) = 0$, as $\boldsymbol{a}_{h-1,\theta} \neq a$. Consequently, $\mathrm{vol}(\mathcal{X}_h(\boldsymbol{a}_{h-1}) \cap \mathcal{P}_\theta(a)) = 0$, and by Equation (14), we have:

$$\mathcal{P}_\theta(a \mid \mathcal{X}_h(\boldsymbol{a}_{h-1})) = \emptyset = \mathcal{P}'_\theta(a \mid \mathcal{X}_h(\boldsymbol{a}_{h-1})).$$

As a result, Algorithm 3 computes $\mathcal{P}_\theta(\cdot|\mathcal{X}_h(\cdot))$ correctly for every $\theta \in \widetilde{\Theta}_h$, concluding the proof. $\square$

**Lemma C.5.** *If Condition C.1 is satisfied and every execution of `Stackelberg` is successful, Algorithm 3 correctly computes $\mathcal{Y}_h$ according to Equation (14).*

*Proof.* We let $\mathcal{Y}_h$ be computed according to Equation (14), and $\mathcal{Y}'_h$ be the one computed by Algorithm 3 at Line 15.

Formally, for every $\boldsymbol{a}_h \in \mathcal{A}_F(\widetilde{\Theta}_h)$ we let $\boldsymbol{a}_{h-1}$ be the restriction of $\boldsymbol{a}_h$ to $\mathcal{A}_F(\widetilde{\Theta}_{h-1})$ and:

$$\mathcal{Y}_h(\boldsymbol{a}_h) := \begin{cases} \mathcal{P}(\boldsymbol{a}_h) \cap \mathcal{X}_h(\boldsymbol{a}_{h-1}) & \text{if its volume is greater than zero} \\ \emptyset & \text{otherwise,} \end{cases}$$

according to Equation (14). As for Lemma C.4, Algorithm 3 correctly computes the regions $\mathcal{P}_\theta(a \mid \mathcal{X}_h(\boldsymbol{a}_{h-1}))$, therefore Line 15 Algorithm 3 computes:

$$\mathcal{Y}'_h(\boldsymbol{a}_h) = \begin{cases} \bigcap_{\theta \in \widetilde{\Theta}_h} \mathcal{P}_\theta(\boldsymbol{a}_{h,\theta} \mid \mathcal{X}_h(\boldsymbol{a}_{h-1})) & \text{if its volume is greater than zero} \\ \emptyset & \text{otherwise.} \end{cases} \tag{15}$$

The two regions $\mathcal{Y}_h(\boldsymbol{a}_h)$ and $\mathcal{Y}'_h(\boldsymbol{a}_h)$ are both empty when $\mathcal{X}_h(\boldsymbol{a}_{h-1}) = \emptyset$. We now show that the two coincide even when $\mathcal{X}_h(\boldsymbol{a}_{h-1}) \neq \emptyset$. Observe that Condition C.1 requires that $\text{vol}(\mathcal{X}_h(\boldsymbol{a}_{h-1})) > 0$ when the polytope is not empty.

Suppose that $\mathcal{P}_\theta(\boldsymbol{a}_{h,\theta} \mid \mathcal{X}_h(\boldsymbol{a}_{h-1}))$ is not empty for every $\theta \in \widetilde{\Theta}_h$. Then, by Equation (13), $\mathcal{P}_\theta(\boldsymbol{a}_{h,\theta} \mid \mathcal{X}_h(\boldsymbol{a}_{h-1})) = \mathcal{P}_\theta(\boldsymbol{a}_{h,\theta}) \cap \mathcal{X}_h(\boldsymbol{a}_{h-1})$. As a result:

$$\mathcal{P}(\boldsymbol{a}_h) \cap \mathcal{X}_h(\boldsymbol{a}_{h-1}) = \bigcap_{\theta \in \widetilde{\Theta}_h} \mathcal{P}_\theta(\boldsymbol{a}_{h,\theta}) \cap \mathcal{X}_h(\boldsymbol{a}_{h-1}) = \bigcap_{\theta \in \widetilde{\Theta}_h} \mathcal{P}_\theta(\boldsymbol{a}_{h,\theta} \mid \mathcal{X}_h(\boldsymbol{a}_{h-1})),$$

where the first equality uses the definition of $\mathcal{P}(\boldsymbol{a}_h)$, and the second Equation (13). Therefore, by applying Equation (14) and Equation (15), we have that $\mathcal{Y}_h(\boldsymbol{a})$ and $\mathcal{Y}'_h(\boldsymbol{a})$ coincide.

Suppose now that there exists some $\bar{\theta} \in \widetilde{\Theta}_h$ such that $\mathcal{P}_{\bar{\theta}}(\boldsymbol{a}_{h,\bar{\theta}} \mid \mathcal{X}_h(\boldsymbol{a}_{h-1}))$ is empty. Then according to Equation (15) $\mathcal{Y}'_h(\boldsymbol{a}_h)$ is also empty. Furthermore, by Equation (13), for $\mathcal{P}_{\bar{\theta}}(\boldsymbol{a}_{h,\bar{\theta}} \mid \mathcal{X}_h(\boldsymbol{a}_{h-1}))$ to be empty, we need $\mathcal{P}_{\bar{\theta}}(\boldsymbol{a}_{h,\bar{\theta}})$ to have null volume (possibly to be empty). This implies that $\mathcal{P}(\boldsymbol{a}_h) \cap \mathcal{X}_h(\boldsymbol{a}_{h-1})$ has zero volume too, as it is a subset of $\mathcal{P}_{\bar{\theta}}(\boldsymbol{a}_{h,\bar{\theta}})$. As a result, $\mathcal{Y}_h(\boldsymbol{a}_h)$ is empty and coincides with $\mathcal{Y}'_h(\boldsymbol{a}_h)$, concluding the proof. $\square$

**Lemma C.3.** *[Formal version of Lemma 4.3] Suppose that Condition C.1 is satisfied and each polytope $\mathcal{X}_h(\boldsymbol{a})$ composing $\mathcal{X}_h$ is defined as the intersections of $N$ hyperplanes whose coefficients have bit complexity bounded by some $B \geq L$. Then Algorithm 3 computes $\mathcal{Y}_h$ according to Equation (14) with probability at least $1 - \delta_2$, using at most:*

$$\widetilde{\mathcal{O}}\left( \frac{1}{\epsilon_h} K^{m+1} n^{2m+2} \left( m^7 B \log^2 \left( \frac{1}{\delta_2} \right) + \binom{N+n}{m} \right) \right)$$

*rounds.*

*Proof.* As of Lemma C.5, Algorithm 3 computes $\mathcal{Y}_h$ correctly if every execution of `Stackelberg` is successful. We therefore have to bound the number of rounds required by these procedures and the probability that they correct terminate.

The `Stackelberg` subprocedure is executed once for every type $\theta \in \widetilde{\Theta}_h \setminus \widetilde{\Theta}_{h-1}$ and every action profile $\boldsymbol{a}_h \in \mathcal{A}_F(\widetilde{\Theta}_{h-1})$. However, it takes exactly zero rounds when it is executed over an empty region $\mathcal{X}_h(\boldsymbol{a}_{h-1})$. By (Personnat et al., 2025) Lemma 3.2 the number of non-empty regions $\mathcal{X}_h(\boldsymbol{a}_{h-1})$ is at most $K^m n^{2m}$. As a result, the number of actual calls to the subrpocedure is bounded by $K^{m+1} n^{2m}$, which accounts for at most $K$ calls for each non-empty region.

According to Lemma C.2, under the event that each query ends in a finite number of rounds, each execution of `Stackelberg` correctly terminates with probability at least $\zeta$ and employs

$$\widetilde{\mathcal{O}}\left( n^2 \left( m^7 B \log \frac{1}{\zeta} + \binom{N+n}{m} \right) \right) = \mathcal{O}\left( n^2 \left( m^7 B \log \frac{1}{\delta_2} + \binom{N+n}{m} \right) \right)$$

queries, where $\zeta := \delta/2$ is defined at Line 2 Algorithm 3. Therefore, the number of queries $C$ performed by Algorithm 3 is:

$$C \leq \widetilde{\mathcal{O}}\left( K^{m+1} n^{2m} n^2 \left( m^7 B \log \frac{1}{\delta_2} + \binom{N+n}{m} \right) \right)$$

with probability at least $1 - \zeta = 1 - \delta_2/2$.

In order to conclude the proof, we bound the number of samples required by the algorithm to terminate correctly with probability at least $1 - \delta$. We observe that every query is performed over a type $\theta$ such that $\mu_\theta \geq \epsilon_h$. A simple probabilistic argument (see *e.g.*, Lemma 2 in (Bacchiocchi et al., 2024a)) shows that given any $\rho \in (0, 1)$, a query ends in at most $T_{\mathrm{q}}(\rho) := \lceil 1/\epsilon \log(1/\rho) \rceil$ with probability at least $1 - \rho$, *i.e.*, with probability at least $1 - \rho$ any given type appears in $T_{\mathrm{q}}(\rho)$ rounds. By considering $\rho = \varsigma/2C$ and employing an union bound over the event that Algorithm 3 executes $C$ queries, we have that with probability at least:

$$1 - \zeta - C\rho = 1 - \frac{\delta_2}{2} - C\frac{\zeta}{2C} = 1 - \delta_2$$

Algorithm 3 terminates correctly in $T_{\mathrm{P}} := CT_{\mathrm{q}}(\rho)$ rounds. In order to conclude the proof, we observe that:

$$
\begin{aligned}
T_{\mathrm{P}} &= CT_{\mathrm{q}} \\
&= \widetilde{\mathcal{O}}\left(\frac{C}{\epsilon_h} \log\left(\frac{C}{\delta_2}\right)\right) \\
&= \widetilde{\mathcal{O}}\left(\frac{1}{\epsilon_h} K^{m+1} n^{2m+2} \left(m^7 B \log^2 \frac{1}{\delta_2} + \binom{N+n}{m}\right)\right),
\end{aligned}
$$

proving the statement. $\qquad\square$

# D. Proofs Omitted from Section 4.3

In this section we provide the formal version of Lemma 4.4 (Lemma D.5) and its proof. Specifically, Lemma 4.4 informally requires that Algorithm 2 and Algorithm 3 were executed "successfully", while Lemma D.5 formalizes the condition that must be satisfied to correctly execute Algorithm 4 as follows.

**Condition D.1.** *Algorithm 4 is called with the following inputs:*

1. *a parameter $\epsilon_h \in (0, 1)$.*

2. *two sets of types $\widetilde{\Theta}_{h-1} \subseteq \widetilde{\Theta}_h \subseteq \Theta$ such that $\widetilde{\Theta}_h \neq \emptyset$ and $\mu_\theta \leq 3\epsilon_h$ for every $\theta \in \Theta \setminus \widetilde{\Theta}_h$.*

3. *a prior estimator $\widehat{\mu}_h$ such that $\|\mu - \widehat{\mu}_h\|_\infty \leq \epsilon_h$.*

4. *a search space $\mathcal{X}_h = \bigcup_{\boldsymbol{a}_{h-1} \in \mathcal{A}_{\mathrm{F}}(\widetilde{\Theta}_{h-1})} \mathcal{X}_h(\boldsymbol{a}_{h-1})$ where each $\mathcal{X}_h(\boldsymbol{a}_{h-1}) \subseteq \mathcal{P}(\boldsymbol{a}_{h-1})$ is either empty or a polytope with volume greater than zero.*

5. *a mapping $\mathcal{Y}_h$ satisfies Equation (14) and such that an optimal commitment $x^\star$ belongs to some $\mathcal{Y}_h(\boldsymbol{a}_h)$, with $\boldsymbol{a}_{h,\theta} = a_\theta^\star(x^\star)$ for every $\theta \in \widetilde{\Theta}_h$.*

The remaining part of the appendix is organized as follows. First, we provide two additional lemmas, namely Lemma D.2 and Lemma D.3. The first provides upper and lower bounds on $\sum_{\theta \in \widetilde{\Theta}_h} \mu_\theta u^{\mathrm{L}}(x, \boldsymbol{a}_\theta)$, which is the leader utility in $x \in \Delta(\mathcal{A}_{\mathrm{L}})$ assuming that only the types in $\widetilde{\Theta}_h$ are drawn from $\mu$, and that they respond according to the action profile $\boldsymbol{a}_\theta \in \mathcal{A}_{\mathrm{F}}(\widetilde{\Theta}_h)$. Lemma D.3 provides instead upper and lower bounds on the actual utility $u^{\mathrm{L}}(x)$ in the terms of the estimated utility $\widehat{u}_h^{\mathrm{L}}(x, \boldsymbol{a}_h)$, for any $\boldsymbol{a}_h \in \mathcal{A}_{\mathrm{F}}(\widetilde{\Theta}_h)$ and $x \in \mathcal{Y}_h(\boldsymbol{a}_h)$. Together, these two lemmas are employed to prove Lemma D.5. Finally, we conclude this appendix with Lemma D.6 and its proof, which will be instrumental to upper bound the number of facets of each polytope composing $\mathcal{X}_h$ (see Lemma E.6).

**Lemma D.2.** *If Condition D.1 is satisfied, then for every $\boldsymbol{a} \in \mathcal{A}_{\mathrm{F}}(\widetilde{\Theta}_h)$ and $x \in \Delta(\mathcal{A}_{\mathrm{L}})$, we have:*

$$\widehat{u}_h^{\mathrm{L}}(x, \boldsymbol{a}) - K\epsilon \leq \sum_{\theta \in \widetilde{\Theta}_h} \mu_\theta u^{\mathrm{L}}(x, \boldsymbol{a}_\theta) \leq \widehat{u}_h^{\mathrm{L}}(x, \boldsymbol{a}) + K\epsilon.$$

*Proof.* We observe that:

$$\left| \widehat{u}_h^{\mathrm{L}}(x, \boldsymbol{a}) - \sum_{\theta \in \widetilde{\Theta}_h} \mu_\theta u^{\mathrm{L}}(x, \boldsymbol{a}_\theta) \right| = \left| \sum_{\theta \in \widetilde{\Theta}_h} (\widehat{\mu}_{h\theta} - \mu_\theta) u^{\mathrm{L}}(x, \boldsymbol{a}_\theta) \right| \leq K\epsilon_h$$

where the inequality holds because, under Condition D.1, $\|\widehat{\mu}_h - \mu\|_\infty \le \epsilon$ and $|\widetilde{\Theta}_h| \le K$. The statement follows by unraveling the absolute value. $\qquad\square$

**Lemma D.3.** *If Condition D.1 is satisfied, then for every $\boldsymbol{a}_h \in \mathcal{A}_\mathrm{F}(\widetilde{\Theta}_h)$ and $x \in \mathcal{Y}_h(\boldsymbol{a}_h)$:*

$$u^\mathrm{L}(x) \ge \widehat{u}_h^\mathrm{L}(x, \boldsymbol{a}_h) - \epsilon |\widetilde{\Theta}_h|.$$

*Furthermore, if $\boldsymbol{a}_{h,\theta} = a_\theta^\star(x)$ for every $\theta \in \widetilde{\Theta}_h$, it holds that:*

$$u^\mathrm{L}(x) \le \widehat{u}_h^\mathrm{L}(x, \boldsymbol{a}_h) + 4K\epsilon_h.$$

*Proof.* Consider any $\boldsymbol{a}_h \in \mathcal{A}_\mathrm{F}(\widetilde{\Theta}_h)$ and let $\boldsymbol{a}_{h-1}$ be its restriction to $\mathcal{A}_\mathrm{F}(\widetilde{\Theta}_{h-1})$. Thanks to Equation (14), we have that $\mathcal{Y}_h(\boldsymbol{a}_h) \subseteq \mathcal{P}(\boldsymbol{a}_h)$. Now take any $x \in \mathcal{Y}_h(\boldsymbol{a}_h) \subseteq \mathcal{P}(\boldsymbol{a}_h)$. We observe that for every $\theta \in \widetilde{\Theta}_h$, a follower of type $\theta$ is indifferent in $x$ between action $\boldsymbol{a}_{h,\theta}$ and $a_\theta^\star(x)$.[8] As the follower breaks ties in favor of the leader, we have:

$$u^\mathrm{L}(x, a_\theta^\star(x)) \ge u^\mathrm{L}(x, \boldsymbol{a}_{h,\theta}). \tag{16}$$

By employing this inequality, we lower bound the leader's utility in $x$ as follows:

$$
\begin{aligned}
u^\mathrm{L}(x) &= \sum_{\theta \in \Theta} \mu_\theta u^\mathrm{L}(x, a_\theta^\star(x)) \\
&= \sum_{\theta \in \widetilde{\Theta}_h} \mu_\theta u^\mathrm{L}(x, a_\theta^\star(x)) + \sum_{\theta \notin \widetilde{\Theta}_h} \mu_\theta u^\mathrm{L}(x, a_\theta^\star(x)) \\
&\ge \sum_{\theta \in \widetilde{\Theta}_h} \mu_\theta u^\mathrm{L}(x, \boldsymbol{a}_{h,\theta}) + \sum_{\theta \notin \widetilde{\Theta}_h} \mu_\theta u^\mathrm{L}(x, a_\theta^\star(x)) \\
&\ge \sum_{\theta \in \widetilde{\Theta}_h} \mu_\theta u^\mathrm{L}(x, \boldsymbol{a}_{h,\theta}) \\
&= \sum_{\theta \in \widetilde{\Theta}_h} \widehat{\mu}_{h,\theta} u^\mathrm{L}(x, \boldsymbol{a}_{h,\theta}) + \sum_{\theta \in \widetilde{\Theta}_h} (\mu_\theta - \widehat{\mu}_{h,\theta}) u^\mathrm{L}(x, \boldsymbol{a}_\theta) \\
&\ge \sum_{\theta \in \widetilde{\Theta}_h} \widehat{\mu}_{h,\theta} u^\mathrm{L}(x, \boldsymbol{a}_{h,\theta}) - \epsilon_h |\widetilde{\Theta}_h| = \widehat{u}_h^\mathrm{L}(x, \boldsymbol{a}_h) - \epsilon_h |\widetilde{\Theta}_h|
\end{aligned}
$$

where the first inequality follows from Equation (16), the second removes a non-negative quantity, and the last one follows from Condition D.1, in particular that $\|\mu - \widehat{\mu}_h\|_\infty \le \epsilon_h$ and that the utility is bounded in $[0, 1]$.

Now consider an action profile $\boldsymbol{a}_h \in \mathcal{A}_\mathrm{F}(\widetilde{\Theta}_h)$ a commitment $x \in \mathcal{Y}_h(\boldsymbol{a}_h)$ such that $\boldsymbol{a}_{h,\theta} = a_\theta^\star(x)$ for every $\theta \in \widetilde{\Theta}_h$. We can upper bound $u^\mathrm{L}(x)$ as follows:

$$
\begin{aligned}
u^\mathrm{L}(x) &= \sum_{\theta \in \Theta} \mu_\theta u^\mathrm{L}(x, a_\theta^\star(x)) \\
&= \sum_{\theta \in \widetilde{\Theta}_h} \mu_\theta u^\mathrm{L}(x, \boldsymbol{a}_{h,\theta}) + \sum_{\theta \in \Theta \setminus \widetilde{\Theta}_h} \mu_\theta u^\mathrm{L}(x, a_\theta^\star(x)) \\
&\le \sum_{\theta \in \widetilde{\Theta}_h} \mu_\theta u^\mathrm{L}(x, \boldsymbol{a}_{h,\theta}) + 3K\epsilon_h \\
&= \sum_{\theta \in \widetilde{\Theta}_h} \widehat{\mu}_{h,\theta} u^\mathrm{L}(x, \boldsymbol{a}_{h,\theta}) + \sum_{\theta \in \widetilde{\Theta}_h} (\mu_\theta - \widehat{\mu}_{h,\theta}) u^\mathrm{L}(x, \boldsymbol{a}_{h,\theta}) + 3K\epsilon_h \\
&\le \sum_{\theta \in \widetilde{\Theta}_h} \widehat{\mu}_{h,\theta} u^\mathrm{L}(x, \boldsymbol{a}_{h,\theta}) + K\epsilon_h + 3K\epsilon_h = \widehat{u}_h^\mathrm{L}(x, \boldsymbol{a}_h) + 4K\epsilon_h.
\end{aligned}
$$

The first inequality follows from the fact that $\mu_\theta \le 3\epsilon_h$ for $\theta \notin \widetilde{\Theta}_h$, while the second one applies $\|\mu - \widehat{\mu}_h\|_\infty \le \epsilon_h$. Both properties are guaranteed by Condition D.1. $\qquad\square$

---

[8] Notice that the two actions are different when $x$ is on the boundary between two best-response regions.

**Lemma D.4.** *If Condition D.1 is satisfied, then Algorithm 4 computes* $\underline{\mathrm{OPT}}_h$ *such that:*

$$\mathrm{OPT} - K\epsilon_h(4 + C_2) \leq \underline{\mathrm{OPT}}_h \leq \mathrm{OPT} - K\epsilon(C_2 - 1),$$

*where* $C_1$ *and* $C_2$ *are computed at Line 1 in Algorithm 3.*

*Proof.* Let $x^\star \in \mathcal{X}_h$ be an optimal commitment which, thanks to Condition D.1, belongs to some $\mathcal{Y}_h(\boldsymbol{a}^\star), \boldsymbol{a}^\star \in \mathcal{A}_{\mathrm{F}}(\widetilde{\Theta}_h)$ such that $\boldsymbol{a}_\theta^\star = a_\theta^\star(x^\star)$ for every $\theta \in \widetilde{\Theta}_h$. Then we lower bound $\underline{\mathrm{OPT}}_h$ as follows:

$$\begin{aligned}
\underline{\mathrm{OPT}}_h &= \max_{\substack{\boldsymbol{a} \in \mathcal{A}_{\mathrm{F}}(\widetilde{\Theta}_h) \\ x \in \mathcal{Y}_h(\boldsymbol{a})}} \widehat{u}_h^{\mathrm{L}}(x, \boldsymbol{a}) - C_2 K \epsilon_h \\
&\geq \widehat{u}_h^{\mathrm{L}}(x^\star, \boldsymbol{a}^\star) - C_2 K \epsilon_h \\
&\geq u^{\mathrm{L}}(x^\star) - K\epsilon_h(4 + C_2),
\end{aligned}$$

where the first inequality holds by the $\max$ operator, and the last inequality by Lemma D.3.

To provide an upper bound, let:

$$x^\circ, \boldsymbol{a}^\circ \in \arg\max_{\substack{\boldsymbol{a} \in \mathcal{A}_{\mathrm{F}}(\widetilde{\Theta}_h) \\ x \in \mathcal{Y}_h(\boldsymbol{a})}} \widehat{u}_h^{\mathrm{L}}(x, \boldsymbol{a}).$$

Then:

$$\begin{aligned}
\underline{\mathrm{OPT}}_h &= \max_{\substack{\boldsymbol{a} \in \mathcal{A}_{\mathrm{F}}(\widetilde{\Theta}_h) \\ x \in \mathcal{Y}_h(\boldsymbol{a})}} \widehat{u}_h^{\mathrm{L}}(x, \boldsymbol{a}) - C_2 K \epsilon_h \\
&= \widehat{u}_h^{\mathrm{L}}(x^\circ, \boldsymbol{a}^\circ) - C_2 K \epsilon_h \\
&\leq u^{\mathrm{L}}(x^\circ) + K\epsilon_h - C_2 K \epsilon_h \\
&\leq \mathrm{OPT} - K\epsilon_h(C_2 - 1),
\end{aligned}$$

where the first inequality comes from Lemma D.3 and the last one by the optimality of OPT.  $\square$

**Lemma D.5.** *If Condition D.1 is satisfied, then Algorithm 4 computes a union of polytopes* $\mathcal{X}_{h+1} = \bigcup_{\boldsymbol{a} \in \mathcal{A}_{\mathrm{F}}(\widetilde{\Theta}_h)} \mathcal{X}_{h+1}(\boldsymbol{a})$ *such that* $\mathcal{X}_{h+1}(\boldsymbol{a}) \subseteq \mathcal{P}(\boldsymbol{a})$ *and* $u^{\mathrm{L}}(x) \geq \mathrm{OPT} - K\epsilon_h(5 + C_1 + C_2)$ *for every* $x \in \mathcal{X}_{h+1}$*, where* $C_1$ *and* $C_2$ *are defined at Line 1 in Algorithm 4. Furthermore, there exists an optimal commitment* $x^\star$ *and an action profile* $\boldsymbol{a}_h \in \mathcal{A}_{\mathrm{F}}(\widetilde{\Theta}_h)$ *such that* $x^\star \in \mathcal{X}_{h+1}(\boldsymbol{a}_h)$ *and* $a_\theta^\star(x^\star) = \boldsymbol{a}_{h,\theta}$ *for every* $\theta \in \widetilde{\Theta}_h$

*Proof.* We observe that the regions $\mathcal{Y}_h(\boldsymbol{a})$ are subsets of $\mathcal{P}(\boldsymbol{a})$ by Equation (14), which is verified according to Condition D.1. Therefore, the regions $\mathcal{X}_{h+1}(\boldsymbol{a}) \subseteq \mathcal{Y}_h(\boldsymbol{a})$ computed at Line 5 are subsets of $\mathcal{P}(\boldsymbol{a})$ themselves. To conclude the proof, we analyze the utility of the leader in the commitments $x \in \mathcal{X}_{h+1}$.

Consider a commitment $x \in \mathcal{X}_{h+1}(\boldsymbol{a}_h)$ for some $\boldsymbol{a}_h \in \mathcal{A}_{\mathrm{F}}(\widetilde{\Theta}_h)$. As it is not empty, by construction $\mathcal{X}_{h+1}(\boldsymbol{a}_h) = \mathcal{Y}_h(\boldsymbol{a}_h) \cap \mathcal{H}_h(\boldsymbol{a}_h)$, where $\mathcal{H}_h(\boldsymbol{a}_h)$ is the half-space defined by:

$$\widehat{u}_h^{\mathrm{L}}(x, \boldsymbol{a}_h) + C_1 \epsilon \geq \underline{\mathrm{OPT}}_h.$$

Since $\underline{\mathrm{OPT}}_h \geq \mathrm{OPT} - K\epsilon_h(4 + C_2)$ by Lemma D.4, we have:

$$\widehat{u}_h^{\mathrm{L}}(x, \boldsymbol{a}_h) \geq \mathrm{OPT} - K\epsilon_h(4 + C_1 + C_2).$$

Finally, we apply Lemma D.3 to get:

$$u^{\mathrm{L}}(x) \geq \widehat{u}_h^{\mathrm{L}}(x, \boldsymbol{a}_h) - K\epsilon_h \geq \mathrm{OPT} - K\epsilon_h(5 + C_1 + C_2),$$

proving the lower bound on the utility of every commitment $x \in \mathcal{X}_{h+1}$.

According to Condition D.1, there exists an optimal commitment $x^\star$ belonging to some $\mathcal{Y}_h(\boldsymbol{a}_h)$, with $a_\theta^\star(x^\star) = \boldsymbol{a}_{h,\theta}$ for every $\theta \in \widetilde{\Theta}_h$. To conclude the proof, we show that $x^\star \in \mathcal{X}_{h+1}(\boldsymbol{a}_h)$ and that $\mathrm{vol}(\mathcal{X}_{h+1}(\boldsymbol{a}_h)) > 0$. By construction, Algorithm 4 computes:

$$\mathcal{X}_{h+1}(\boldsymbol{a}_h) = \begin{cases} \mathcal{Y}_h(\boldsymbol{a}_h) \cap \mathcal{H}_h(\boldsymbol{a}_h) & \text{if its volume is greater than zero} \\ \emptyset & \text{otherwise,} \end{cases}$$

where $\mathcal{H}_h(\boldsymbol{a}_h)$ is the half-space computed at Line 4. We therefore have to show that $x^\star \in \mathcal{Y}_h(\boldsymbol{a}_h) \cap \mathcal{H}_h(\boldsymbol{a}_h)$, *i.e.*, $x^\star \in \mathcal{H}_h(\boldsymbol{a}_h)$, and that $\mathcal{Y}_h(\boldsymbol{a}_h) \cap \mathcal{H}_h(\boldsymbol{a}_h)$ has volume larger than zero. Let us observe that by the definition of $\mathcal{H}_h(\boldsymbol{a}_h)$ (Line 4), a point $x \in \mathcal{Y}_h(\boldsymbol{a}_h)$ belongs to $\mathcal{Y}_h(\boldsymbol{a}_h) \cap \mathcal{H}_h(\boldsymbol{a}_h)$ if

$$\widehat{u}_h^{\mathrm{L}}(x, \boldsymbol{a}_h) + C_1 \epsilon_h \geq \underline{\mathrm{OPT}}_h,$$

where $\underline{\mathrm{OPT}}_h \leq \mathrm{OPT} - K\epsilon_h(C_2 - 1)$ according to Lemma D.4. Therefore, for $x$ to belong to $\mathcal{Y}_h(\boldsymbol{a}_h) \cap \mathcal{H}_h(\boldsymbol{a}_h)$ is sufficient that:

$$\widehat{u}_h^{\mathrm{L}}(x, \boldsymbol{a}_h) \geq \mathrm{OPT} - K\epsilon_h(C_1 + C_2 - 1) \tag{17}$$

We first show that $x^\star \in \mathcal{H}_h(\boldsymbol{a}_h)$. By Lemma D.3, we have $\widehat{u}^{\mathrm{L}}(x^\star, \boldsymbol{a}_h) \geq \mathrm{OPT} - 4K\epsilon_h$. Therefore, Equation (17) is satisfied and $x^\star \in \mathcal{H}_h(\boldsymbol{a}_h)$ as long as $C_1 + C_2 \geq 5$.

In order to complete the proof, we have to show that $\mathcal{Y}_h(\boldsymbol{a}_h) \cap \mathcal{H}_h(\boldsymbol{a}_h)$ has volume larger than zero. To do so, by Lemma F.2 it is sufficient to find a commitment $x^\circ \in \mathcal{H}_h(\boldsymbol{a}_h) \cap \mathrm{interior}(\mathcal{Y}_h(\boldsymbol{a}_h))$.[9] If $x^\star \in \mathrm{interior}(\mathcal{Y}_h(\boldsymbol{a}_h))$, this is satisfied. Suppose instead that $x^\star \in \partial\mathcal{Y}_h(\boldsymbol{a}_h)$. Recall that since $\mathcal{Y}_h(\boldsymbol{a}_h)$ is non-empty, it also has non-zero volume (Equation (14) holds by Condition D.1). Therefore, there exists some $x' \in \mathrm{interior}(\mathcal{Y}_h(\boldsymbol{a}_h))$. Let

$$y := \max(\widehat{u}_h^{\mathrm{L}}(x', \boldsymbol{a}_h), \mathrm{OPT} - 4K\epsilon_h).$$

By Lemma F.1, there exists a point $x^\circ$ in the segment between $x^\star$ and $x'$ with estimated utility $\widehat{u}^{\mathrm{L}}(x^\circ, \boldsymbol{a}_h) = y$. This point is also different from $x^\star$ (if $\widehat{u}^{\mathrm{L}}(x^\star, \boldsymbol{a}_h) = y$, by Lemma F.1 we can take any other point on the segment). As a result, it must be an interior point of $\mathcal{Y}_h(\boldsymbol{a}_h)$. At the same time, $x^\circ$ satisfies Equation (17), and thus belongs to $\mathcal{H}_h(\boldsymbol{a}_h)$. By Lemma F.2 the polytope $\mathcal{Y}_h(\boldsymbol{a}_h) \cap \mathcal{H}_h(\boldsymbol{a}_h)$ has non-zero volume, concluding the proof. $\qquad\square$

**Lemma D.6.** *Suppose that Condition C.1 is satisfied for two successive epochs $h - 1, h$, and that $\widetilde{\Theta}_h = \widetilde{\Theta}_{h-1} =: \widetilde{\Theta}$. Let $\boldsymbol{a} \in \mathcal{A}_\mathrm{F}(\widetilde{\Theta})$ and $\mathcal{H}_{h-1}(\boldsymbol{a}), \mathcal{H}_h(\boldsymbol{a})$ be the half-spaces computed at Line 4 when Algorithm 4 is executed at epochs $h - 1$ and $h$, respectively. Then $\mathcal{H}_h(\boldsymbol{a}) \cap \Delta(\mathcal{A}_\mathrm{L}) \subseteq \mathcal{H}_{h-1}(\boldsymbol{a}) \cap \Delta(\mathcal{A}_\mathrm{L})$.*

*Proof.* Fix some $\boldsymbol{a} \in \mathcal{A}_\mathrm{F}(\widetilde{\Theta})$ such that $\mathcal{X}_{h+1}(\boldsymbol{a}) \neq \emptyset$. Observe that we drop the subscript $h$ from $\boldsymbol{a}$, as $\widetilde{\Theta} = \widetilde{\Theta}_h = \widetilde{\Theta}_{h+1}$.

In order to prove the statement, we show that:

$$\Delta(\mathcal{A}_\mathrm{L}) \cap \mathcal{H}_{h-1}(\boldsymbol{a}) \cap \mathcal{H}_h(\boldsymbol{a}) \supseteq \Delta(\mathcal{A}_\mathrm{L}) \cap \mathcal{H}_h(\boldsymbol{a}). \tag{18}$$

Take any $x \in \Delta(\mathcal{A}_\mathrm{L}) \cap \mathcal{H}_h(\boldsymbol{a})$. To prove Equation (18), we need to show that $x \in \mathcal{H}_{h-1}(\boldsymbol{a})$, that is:

$$U_{h-1} := \widehat{u}_{h-1}^{\mathrm{L}}(x, \boldsymbol{a}) + 2C_1\epsilon_h \geq \underline{\mathrm{OPT}}_{h-1},$$

where we considered that $\epsilon_{h-1} = 2\epsilon_h$. As $x \in \mathcal{H}_h(\boldsymbol{a})$, it holds that:

$$U_h := \widehat{u}_h^{\mathrm{L}}(x, \boldsymbol{a}) + C_1\epsilon_h \geq \underline{\mathrm{OPT}}_h.$$

Therefore, we can prove Equation (18) by showing that $U_h \geq U_{h-1}$ and $\underline{\mathrm{OPT}}_h \leq \underline{\mathrm{OPT}}_{h-1}$. Employing Lemma D.4 to epochs $h$ and $h - 1$ we get:

$$\underline{\mathrm{OPT}}_h \geq \mathrm{OPT} - K\epsilon_h(4 + C_2)$$
$$\underline{\mathrm{OPT}}_{h-1} \leq \mathrm{OPT} - K\epsilon_h(2C_2 - 2).$$

---

[9]We let $\mathrm{interior}(\mathcal{P}) := \mathcal{P} \setminus \partial\mathcal{P}$ be the interior of any given polytope $\mathcal{P}$.

By taking $C_2 \geq 6$ we get $\underline{\mathrm{OPT}}_h \leq \underline{\mathrm{OPT}}_{h-1}$. At the same time, we can bound $U_h$ and $U_{h-1}$ by means of Lemma D.2. Observe that this lemma holds for every $x \in \Delta(\mathcal{A}_\mathrm{L})$. Let $\alpha := \sum_{\theta \in \widetilde{\Theta}} \mu_\theta u^\mathrm{L}(x, \boldsymbol{a}_\theta) \geq 0$. We have:

$$U_h = \widehat{u}_h^\mathrm{L}(x, \boldsymbol{a}) + C_1 \epsilon_h \leq \alpha + K \epsilon_h + C_1 \epsilon_h = \alpha + K \epsilon_h (C_1 + 1)$$
$$U_{h-1} = \widehat{u}_{h-1}^\mathrm{L}(x, \boldsymbol{a}) + 2C_1 \epsilon_h \geq \alpha - 2K \epsilon_h + 2C_1 \epsilon_h = \alpha + K \epsilon_h (2C_1 - 2)$$

where we leverage the fact that $\epsilon_{h-1} = 2\epsilon_h$. By taking $C_1 \geq 3$, we have $U_h \leq U_{h-1}$. As a result, Equation (18) holds when $C_1 \geq 3$ and $C_2 \geq 6$, proving the statement. $\qquad\square$

# E. Proofs of the Regret Bound

We first define the following event.

**Definition E.1.** We let $\mathcal{E}_h$ be the event under which, for every epoch $h' \leq h$, Condition C.1 is verified when Algorithm 3 is executed, and Condition D.1 is verified when Algorithm 4 is executed. Furthermore, Algorithm 3 is executed in the number of rounds specified in Lemma C.3.

**Lemma E.2.** *The probability of event $\mathcal{E}_1$ is at least $1 - \delta_1 - \delta_2$.*

*Proof.* We first observe that

$$\Delta(\mathcal{A}_\mathrm{L}) = \mathcal{X}_1 = \bigcup_{\boldsymbol{a} \in \mathcal{A}_\mathrm{F}(\widetilde{\Theta}_0)} \mathcal{X}_1(\boldsymbol{a}),$$

where $\mathcal{A}_\mathrm{F}(\widetilde{\Theta}_0) := \{\perp\}$ and $\mathcal{X}_1(\perp) = \mathcal{P}(\perp) = \Delta(\mathcal{A}_\mathrm{L})$. Consequently, the search space satisfies the requirements of Condition C.1 and Condition D.1. The set $\widetilde{\Theta}_1 \subseteq \Theta$ and the estimator $\widehat{\mu}_1$ are computed by Algorithm 2. As of Lemma 4.1, with probability at least $1 - \delta_1$ both $\widetilde{\Theta}_1$ and $\widehat{\mu}_1$ are computed according to Condition C.1 and Condition D.1. Observe that $\widetilde{\Theta}_1 \neq \emptyset$, as at least one type appears with probability at least $\epsilon_1 = 1/K$. Putting all together, Condition C.1 holds with probability at least $1 - \delta_1$.

We can now apply Lemma C.3 and a union bound, proving that $\mathcal{Y}_1$ satisfies Equation (14) and all the properties above hold. To conclude the proof, we need to show that there exists some $\boldsymbol{a}_1 \in \mathcal{A}_\mathrm{F}(\widetilde{\Theta}_1)$ such that $x^\star \in \mathcal{Y}_1(\boldsymbol{a}_1)$ and $a_\theta^\star(x^\star) = \boldsymbol{a}_{1,\theta}$ for every $\theta \in \widetilde{\Theta}_1$, which implies that Condition D.1 is satisfied. Let $\boldsymbol{a}^\star := (a_\theta^\star(x^\star))_{\theta \in \widetilde{\Theta}_1}$ and $\boldsymbol{a}_1 := \boldsymbol{a}^\star|\widetilde{\Theta}_1$. It suffices to show that $x^\star \in \mathcal{Y}_1(\boldsymbol{a}_1)$. We recall that $\mathrm{vol}(\mathcal{P}(\boldsymbol{a}^\star)) > 0$. Therefore, $\mathrm{vol}(\mathcal{P}(\boldsymbol{a}_1)) > 0$ and $x^\star \in \mathcal{P}(\boldsymbol{a}_1)$, as $\mathcal{P}(\boldsymbol{a}_1) \supseteq \mathcal{P}(\boldsymbol{a}^\star)$. By Equation (14), we have:

$$\mathcal{Y}_1(\boldsymbol{a}_1) = \mathcal{P}(\boldsymbol{a}_1) \cap \mathcal{X}_1(\boldsymbol{a}_1|\widetilde{\Theta}_0) = \mathcal{P}(\boldsymbol{a}_1) \cap \Delta(\mathcal{A}_\mathrm{L}) = \mathcal{P}(\boldsymbol{a}_1).$$

As a result, $x^\star \in \mathcal{Y}_1(\boldsymbol{a}_1)$, concluding the proof. $\qquad\square$

**Lemma E.3.** *With probability at least $1 - H(\delta_1 + \delta_2)$, the event $\mathcal{E}_H$ holds, where $H$ is the number of epochs of Algorithm 1.*

*Proof.* We prove by induction that for every epoch $h \in \{1, \dots, H\}$, it holds

$$\mathbb{P}(\mathcal{E}_h \mid \mathcal{E}_{h-1}) \geq 1 - \delta_1 - \delta_2,$$

where we define $\mathbb{P}(\mathcal{E}_0) := 0$.

The base step $h = 1$ is proved by Lemma E.2. Now suppose that we are under the event $\mathcal{E}_{h-1}$ for any $2 \leq h \leq H$. For the sake of explanation, suppose that the epoch is completed without reaching $T$ rounds.

We now show that with probability at least $1 - \delta_1$, the set $\widetilde{\Theta}_h$ and the estimator $\widehat{\mu}_h$ satisfy the constraints imposed by Condition C.1 and Condition D.1. By Lemma 4.1, the estimator satisfies $\|\mu - \widehat{\mu}_h\|_\infty \leq \epsilon_h$, and the set of types $\bar{\Theta}_h$ is such that $\mu_\theta \geq \epsilon_h$ for each $\theta \in \bar{\Theta}_h$ and $\mu_\theta \leq 3\epsilon_h$ for each $\theta \in \Theta \setminus \bar{\Theta}_h$. Algorithm 1 computes $\widetilde{\Theta}_h := \widetilde{\Theta}_{h-1} \cup \bar{\Theta}_h$. By the inductive hypothesis, $\mu_\theta \geq \epsilon_{h-1} \geq \epsilon_h$ for every $\theta \in \widetilde{\Theta}_{h-1}$, hence $\mu_\theta \geq \epsilon_h$ for every $\theta \in \widetilde{\Theta}_h$. Moreover, every $\theta \notin \widetilde{\Theta}_h$ appears with probability at most $3\epsilon_h$, as $\theta \notin \bar{\Theta}_h$ by construction. Finally, $\widetilde{\Theta}_h \supseteq \widetilde{\Theta}_{h-1}$ is non-empty by the inductive hypothesis. Therefore, both the set of types and the estimator are computed correctly with probability at least $1 - \delta_1$. By Lemma C.3 and a union bound, we also have that $\mathcal{Y}$ satisfies Equation (14) with probability at lest $1 - \delta_1 - \delta_2$.

We now observe that by Lemma D.5 applied to the previous epoch, the search space $\mathcal{X}_h$ is a union of polytopes as required by Condition C.1 and Condition D.1. Furthermore, combining Equation (14) with the result provided by Lemma D.5, one can verify that an optimal commitment belongs $\mathcal{X}_h$ as required by Condition D.1. As a result, with probability $1 - \delta_1 - \delta_2$ both properties hold before the respective algorithms are executed. Hence, $\mathbb{P}(\mathcal{E}_h \mid \mathcal{E}_{h-1}) \geq 1 - \delta_1 - \delta_2$ for every epoch $h \in \{1, \ldots, H\}$. A recursive argument completes the proof. $\square$

**Lemma 4.6.** *The largest epoch index $H \in \mathbb{N}$ executed by Algorithm 1 satisfies $H \leq H' := \log_4(5T)$.*

*Proof.* We observe that to complete epoch $h \in \{1, \ldots, H\}$, Algorithm 1 employs at least $1/\epsilon_h^2$ rounds (see Algorithm 2). Since $\epsilon_h = 1/(K2^h)$, we have that:

$$\sum_{h=1}^{H-1} \frac{1}{\epsilon_h^2} = \sum_{h=1}^{H-1} (K2^h)^2 \leq T,$$

as the rounds to complete $H - 1$ epochs cannot exceed $T$. We do not count epoch $H$ as that the algorithm may not terminate it. We then observe that:

$$T \geq \sum_{h=1}^{H-1} (K2^h)^2 = K^2 \sum_{h=1}^{H-1} 4^h = K^2 \frac{4 - 4^{H+1}}{1 - 4} = K^2 \frac{4^H - 1}{3}.$$

As a result, we have:

$$H \leq \log_4\left(\frac{3T}{K^2} + 4\right) \leq \log_4(5T),$$

concluding the proof. $\square$

To bound the regret of Algorithm, we need to upper bound the number of facets of the polytopes $\mathcal{X}_h(\boldsymbol{a}_{h-1})$, in order to apply Lemma C.3. We define:

$$\Psi := \{\bar{h} \in [H] \mid \widetilde{\Theta}_{\bar{h}} \neq \widetilde{\Theta}_{\bar{h}-1}\} \cup \{H + 1\},$$

where $H$ is the number of epochs. In the following, for the sake of the analysis, we will assume that the last epoch is completed (otherwise it is sufficient to consider a fictitious $\mathcal{X}_{H+1}$ computed as if the algorithm did not terminate after $T$ rounds). We also observe (see Algorithm 4 Line 5) that under the event $\mathcal{E}_H$, for every $h \in [H]$ and $\boldsymbol{a}_{h-1} \in \mathcal{A}_F(\widetilde{\Theta}_{h-1})$ such that $\mathcal{X}_h(\boldsymbol{a}_{h-1}) \neq \emptyset$, we have:

$$\mathcal{X}_h(\boldsymbol{a}_{h-1}) = \mathcal{Y}_{h-1}(\boldsymbol{a}_{h-1}) \cap \mathcal{H}_{h-1}(\boldsymbol{a}_{h-1}), \tag{19}$$

where $\mathcal{H}_{h-1}(\boldsymbol{a}_{h-1})$ is a half-space.

**Lemma E.4.** *Let $\bar{h}, \bar{h}'$ be two successive values in $\Psi$. For every $h \in \{\bar{h} + 1, \ldots, \bar{h}'\}$ and $\boldsymbol{a} \in \mathcal{A}_F(\widetilde{\Theta}_{\bar{h}}) = \mathcal{A}_F(\widetilde{\Theta}_{h-1})$, it holds that $\mathcal{X}_h(\boldsymbol{a}) = \mathcal{Y}_{\bar{h}}(\boldsymbol{a}) \cap \mathcal{H}_{h-1}(\boldsymbol{a})$, unless it is empty.*

*Proof.* The statement is trivially satisfied when $\bar{h} + 1 = \bar{h}' = h$ (see Line 5 Algorithm 4). We therefore assume that

$$\bar{h}' \geq \bar{h} + 2.$$

Let $\boldsymbol{a} \in \mathcal{A}_F(\widetilde{\Theta}_{\bar{h}})$. For every $h \in \{\bar{h} + 1, \ldots, \bar{h}'\}$ such that $\mathcal{X}_h(\boldsymbol{a}) \neq \emptyset$, Algorithm 4 at Line 5 computes:

$$\mathcal{X}_h(\boldsymbol{a}) = \mathcal{Y}_{h-1}(\boldsymbol{a}) \cap \mathcal{H}_{h-1}(\boldsymbol{a}). \tag{20}$$

Furthermore, for every $h \in \{\bar{h} + 1, \ldots, \bar{h}' - 1\}$, we can employ Lemma D.6 as follows:

$$\mathcal{H}_h(\boldsymbol{a}) \cap \Delta(\mathcal{A}_L) \subseteq \mathcal{H}_{h-1}(\boldsymbol{a}) \cap \Delta(\mathcal{A}_L). \tag{21}$$

If $\mathcal{X}_h(\boldsymbol{a}) \neq \emptyset$, we can leverage Equation (14) to get:

$$\mathcal{Y}_h(\boldsymbol{a}) = \mathcal{P}(\boldsymbol{a}) \cap \mathcal{X}_h(\boldsymbol{a}|\widetilde{\Theta}_{h-1}) = \mathcal{P}(\boldsymbol{a}) \cap \mathcal{X}_h(\boldsymbol{a}) = \mathcal{X}_h(\boldsymbol{a}), \tag{22}$$

where the second equality exploit the fact that $\widetilde{\Theta}_{h-1} = \widetilde{\Theta}_h$, and the last equality holds because $\mathcal{X}_h(\boldsymbol{a}) \subseteq \mathcal{P}(\boldsymbol{a})$ under $\mathcal{E}_h$.

We prove by induction that for every $h \in \{\bar{h}+1, \ldots, \bar{h}'\}$ and $\boldsymbol{a} \in \widetilde{\Theta}_{\bar{h}}$ such that $\mathcal{X}_h(\boldsymbol{a}) \neq \emptyset$, it holds that $\mathcal{X}_h(\boldsymbol{a}) = \mathcal{Y}_{\bar{h}} \cap \mathcal{H}_{h-1}(\boldsymbol{a})$. The base step is $h = \bar{h}+1$. Here we have:

$$\mathcal{X}_h(\boldsymbol{a}) = \mathcal{Y}_{h-1}(\boldsymbol{a}) \cap \mathcal{H}_{h-1}(\boldsymbol{a}) = \mathcal{Y}_{\bar{h}}(\boldsymbol{a}) \cap \mathcal{H}_{h-1}(\boldsymbol{a}),$$

where we employed Equation (20) and the equality $h - 1 = \bar{h} + 1 - 1 = \bar{h}$. Now consider any $h \in \{\bar{h}+2, \ldots, \bar{h}'\}$, and assume that $\mathcal{X}_{h-1}(\boldsymbol{a}) = \mathcal{Y}_{\bar{h}}(\boldsymbol{a}) \cap \mathcal{H}_{h-2}(\boldsymbol{a})$. We have:

$$\begin{aligned}
\mathcal{X}_h(\boldsymbol{a}) &= \mathcal{Y}_{h-1}(\boldsymbol{a}) \cap \mathcal{H}_{h-1}(\boldsymbol{a}) \\
&= \mathcal{X}_{h-1}(\boldsymbol{a}) \cap \mathcal{H}_{h-1}(\boldsymbol{a}) \\
&= \mathcal{Y}_{\bar{h}}(\boldsymbol{a}) \cap \mathcal{H}_{h-2}(\boldsymbol{a}) \cap \mathcal{H}_{h-1}(\boldsymbol{a}) \\
&= \mathcal{Y}_{\bar{h}}(\boldsymbol{a}) \cap (\mathcal{H}_{h-2}(\boldsymbol{a}) \cap \Delta(\mathcal{A}_{\mathrm{L}})) \cap (\mathcal{H}_{h-1}(\boldsymbol{a}) \cap \Delta(\mathcal{A}_{\mathrm{L}})) \\
&= \mathcal{Y}_{\bar{h}}(\boldsymbol{a}) \cap \mathcal{H}_{h-1}(\boldsymbol{a}) \cap \Delta(\mathcal{A}_{\mathrm{L}}) = \mathcal{Y}_{\bar{h}}(\boldsymbol{a}) \cap \mathcal{H}_{h-1}(\boldsymbol{a}),
\end{aligned}$$

where we employed Equation (20), Equation (22), the inductive step, the identity $\mathcal{Y}_{\bar{h}}(\boldsymbol{a}) \subseteq \Delta(\mathcal{A}_{\mathrm{L}})$, and Equation (21). Therefore the inductive step holds for $h$, concluding the proof. $\qquad\square$

**Lemma E.5.** *Let $h \in \{2, \ldots, H\}$ and $\bar{h} = \max\{h' \in \Psi \mid h' < h\}$. Under the event $\mathcal{E}_H$, every non-empty polytope $\mathcal{X}_h(\boldsymbol{a}_{h-1})$ composing $\mathcal{X}_h$ is defined as:*

$$\mathcal{X}_h(\boldsymbol{a}_{h-1}) = \mathcal{P}(\boldsymbol{a}_{\bar{h}}) \cap \bigcap_{\bar{h}' \in \Psi : 1 < \bar{h}' < \bar{h}} \mathcal{H}_{\bar{h}'-1}(\boldsymbol{a}_{\bar{h}'}) \cap \mathcal{H}_{h-1}(\boldsymbol{a}_{h-1}),$$

*where $\boldsymbol{a}_{\bar{h}'} = \boldsymbol{a}_{h-1}|\widetilde{\Theta}_{\bar{h}'}$.*

*Proof.* In the following we let $\bar{h}^{\diamond} := \min \Psi \setminus \{1\}$.

Let $\bar{h} \in \Psi \setminus \{1, H+1\}$ and $\boldsymbol{a}_{\bar{h}} \in \mathcal{A}_{\mathrm{F}}(\widetilde{\Theta}_{\bar{h}})$ such that $\mathcal{X}_{\bar{h}}(\boldsymbol{a}_{\bar{h}}) \neq \emptyset$. We also let $\bar{h}'$ the previous value in $\Psi$. Thanks to Equation (14) and Lemma E.4, we have:

$$\mathcal{Y}_{\bar{h}}(\boldsymbol{a}_{\bar{h}}) = \mathcal{P}(\boldsymbol{a}_{\bar{h}}) \cap \mathcal{X}_{\bar{h}}(\boldsymbol{a}_{\bar{h}-1}) = \mathcal{P}(\boldsymbol{a}_{\bar{h}}) \cap \mathcal{Y}_{\bar{h}'}(\boldsymbol{a}_{\bar{h}'}) \cap \mathcal{H}_{\bar{h}-1}(\boldsymbol{a}_{\bar{h}'}).$$

Recursively applying this step to $\mathcal{Y}_{\bar{h}'}(\boldsymbol{a}_{\bar{h}'})$ till reaching epoch 1 and considering that $\mathcal{P}(\boldsymbol{a}_{\bar{h}''}) \subseteq \mathcal{P}(\boldsymbol{a}_{\bar{h}})$ for $\bar{h}'' \leq \bar{h}$ and $\mathcal{Y}_1(\boldsymbol{a}_1) = \mathcal{P}(\boldsymbol{a}_1)$, we get:

$$\mathcal{Y}_{\bar{h}}(\boldsymbol{a}_{\bar{h}}) = \mathcal{P}(\boldsymbol{a}_{\bar{h}}) \cap \bigcap_{\bar{h}' \in \Psi : 1 < \bar{h}' < \bar{h}} \mathcal{H}_{\bar{h}'-1}(\boldsymbol{a}_{\bar{h}'}). \tag{23}$$

Now consider an epoch $h \in \{\bar{h}^{\diamond}+1, \ldots, H\}$ and an action profile $\boldsymbol{a}_{h-1} \in \widetilde{\Theta}_{h-1}$ such that $\mathcal{X}_h(\boldsymbol{a}_{h-1}) \neq \emptyset$. There exists $\bar{h} = \max\{\bar{h}' \in \Psi \mid \bar{h}' < h\}$ different from one. We can therefore employ Lemma E.4 and Equation (23) to get:

$$\begin{aligned}
\mathcal{X}_h(\boldsymbol{a}_{h-1}) &= \mathcal{Y}_{\bar{h}}(\boldsymbol{a}_{\bar{h}}) \cap \mathcal{H}_{h-1}(\boldsymbol{a}_{h-1}) \\
&= \mathcal{P}(\boldsymbol{a}_{\bar{h}}) \cap \bigcap_{\bar{h}' \in \Psi : 1 < \bar{h}' < \bar{h}} \mathcal{H}_{\bar{h}'-1}(\boldsymbol{a}_{\bar{h}'}) \cap \mathcal{H}_{h-1}(\boldsymbol{a}_h).
\end{aligned}$$

Consider instead an epoch $h \in \{2, \ldots, \bar{h}^{\diamond}\}$ and let $\bar{h} = 1$. By employing Lemma E.4 and Equation (14) we get:

$$\begin{aligned}
\mathcal{X}_h(\boldsymbol{a}_{h-1}) &= \mathcal{Y}_1(\boldsymbol{a}_{h-1}) \cap \mathcal{H}_{h-1}(\boldsymbol{a}_{h-1}) \\
&= \mathcal{P}(\boldsymbol{a}_{h-1}) \cap \mathcal{H}_{h-1}(\boldsymbol{a}_{h-1}),
\end{aligned}$$

concluding the proof. $\qquad\square$

**Lemma E.6.** *Under the event $\mathcal{E}_H$, whenever Algorithm 3 is executed, every $\mathcal{X}_h(\boldsymbol{a}_{h-1})$ has at most $N \leq Kn+m+K$ facets. Furthermore, the coefficients of the hyperplanes defining these facets can be encoded by at most $\mathcal{O}(L+\log(T)+\log(K)+B_{\delta})$ bits, where $B_{\delta}$ is the bit complexity of the parameter $\delta$ given in input to Algorithm 1.*

*Proof.* For the sake of the analysis, we will assume that the last epoch is completed (otherwise it is sufficient to consider a fictitious $\mathcal{X}_{H+1}$). Given any $\boldsymbol{a}_h \in \mathcal{A}_{\mathrm{F}}(\widetilde{\Theta}_h)$ for some epoch $h$, we let $\boldsymbol{a}_{h'} = \boldsymbol{a}_h | \widetilde{\Theta}_{h'}$ for every $0 \le h' < h$.

We let $N_h^{\mathrm{X}}$ and $N_h^{\mathrm{Y}}$ be the maximum number of facets of any region $\mathcal{X}_h(\boldsymbol{a}_{h-1})$ and $\mathcal{Y}_h(\boldsymbol{a}_h)$ at epoch $h$, respectively. To bound the number of facets of a polytope, we will bound the number of half-spaces defining it. Notice that every polytope that we consider is a subset of the hyperplane containing $\Delta(\mathcal{A}_{\mathrm{L}})$. This hyperplane will not be considered, as it does not define a facet.

Let $h \in \{2, \dots, H\}$ and $\bar{h} = \max\{h' \in \Psi \mid h' < h\}$. Consider a non-empty polytope $\mathcal{X}_h(\boldsymbol{a}_{h-1})$ with $\boldsymbol{a}_{h-1} \in \mathcal{A}_{\mathrm{F}}(\widetilde{\Theta}_{h-1})$. By Lemma E.5 we have:

$$\mathcal{X}_h(\boldsymbol{a}_{h-1}) = \mathcal{P}(\boldsymbol{a}_{\bar{h}}) \cap \bigcap_{\bar{h}' \in \Psi : 1 < \bar{h}' < \bar{h}} \mathcal{H}_{\bar{h}'-1}(\boldsymbol{a}_{\bar{h}'}) \cap \mathcal{H}_{h-1}(\boldsymbol{a}_{h-1}).$$

We observe that $\mathcal{P}(\boldsymbol{a}_{\bar{h}})$ has at most $|\widetilde{\Theta}_{\bar{h}}| n + m$ facets. Furthermore, $|\Psi \setminus \{1, H+1\}| \le K - 1$. The single epoch we did not consider was the first one. Since $\mathcal{X}_1 = \Delta(\mathcal{A}_{\mathrm{L}})$ has $m$ facets, the number of facets is at most $Kn + m + K$ for every epoch $h \in [1, H]$.

To conclude the proof, we need to upper bound the number $B$ of bits required to encode the coefficients of the hyperplanes defying the facets of any $\mathcal{X}_h(\boldsymbol{a}_h)$. For the facets of the region $\mathcal{P}(\boldsymbol{a}_{\bar{h}})$, these coefficients have at most $L$ bits. The hyperplanes $\mathcal{H}_h(\boldsymbol{a}_h)$ are instead defined by the inequality:

$$\sum_{\theta \in \widetilde{\Theta}_h} \widehat{\mu}_{h,\theta} u^{\mathrm{L}}(x, \boldsymbol{a}_\theta) + C_1 K \epsilon_h \ge \max_{\boldsymbol{a} \in \mathcal{A}_{\mathrm{F}}(\widetilde{\Theta}_h)} \max_{x \in \mathcal{Y}_h(\boldsymbol{a})} \sum_{\theta \in \widetilde{\Theta}_h} \widehat{\mu}_{h,\theta} u^{\mathrm{L}}(x, \boldsymbol{a}_\theta) - C_2 K \epsilon_h,$$

where $\epsilon_h = 1/(K 2^h)$ and $h \le H \le \log_4(5T)$ as of Lemma 4.6. Therefore, $\epsilon_h$ can be represented by

$$\mathcal{O}(\log(T) + \log(K))$$

bits. Similarly, $C K \epsilon_h$ can be represented by $\mathcal{O}(\log(T) + \log(K))$ bits for any constant $C$. The prior estimator $\widehat{\mu}_h$ has instead been computed by Algorithm 2 as the empirical estimator of $\mu$ using $\mathcal{O}(1/\epsilon_h^2 \log(K/\delta_1))$ samples, where $\delta_1$ defined at Line 2 Algorithm 1 has bit complexity bounded by $\mathcal{O}(\log(\log(T)) + B_\delta)$. Therefore, each component of $\widehat{\mu}_h$ can be represented by at most

$$\mathcal{O}(\log(T) + \log(K) + B_\delta).$$

Overall, each coefficient of the hyperplane can be encoded by at most $\mathcal{O}(L + \log(T) + \log(K) + B_\delta)$, accounting for $L$ bits to represent the leader utility. As a result, $B \le \mathcal{O}(L + \log(T) + \log(K) + B_\delta)$. $\qquad\square$

**Theorem E.7.** *With probability at least $1 - \delta$, the regret of Algorithm 1 is:*

$$R_T \le \widetilde{\mathcal{O}}\left(K^2 \log\left(\frac{K}{\delta}\right) \sqrt{T} + \beta \log^2(T)\right),$$

*where:*

$$\beta := K^{m+2} n^{2m+2} \left(m^7 (L + B_\delta) \log\left(\frac{1}{\delta}\right) + (Kn + m)^m\right)$$

*is a time-independent function of the instance size and $\delta$.*

*Proof.* By Lemma E.3 the event $\mathcal{E}_H$ happens with probability at least $1 - H(\delta_1 + \delta_2)$. We also have $H \le \log_4(5T)$ by Lemma 4.6. By taking

$$\delta_1 = \delta_2 = \frac{\delta}{2 \lceil \log_4(5T) \rceil},$$

the clean event holds with probability at least $1 - \delta$. Let us observe that:

$$\frac{1}{\delta'} = \mathcal{O}\left(\frac{1}{\delta} \log(T)\right).$$

Now we bound the regret of Algorithm 1 under the event $\mathcal{E}_H$. We will let $B := L + B_\delta$ and $N := Kn + m + K$. We also define the following two quantities:

$$\alpha_1 := \log\left(\frac{K}{\delta}\right)$$

$$\alpha_2 := K^{m+1}n^{2m+2}\left(m^7 B \log\left(\frac{1}{\delta}\right) + \binom{N+n}{m}\right).$$

Consider an epoch $h \in \{1, \ldots, H\}$. The number of rounds of epoch $h$ is bounded by $T_h := T_{h,1} + T_{h,2}$, where $T_{h,1}$ and $T_{h,2}$ are the number of rounds to execute Algorithm 2 and Algorithm 3, respectively. By Lemma 4.1 we have:

$$T_{h,1} \le \frac{1}{\epsilon_h^2}\log\left(\frac{K}{\delta'}\right) = \mathcal{O}\left(\frac{1}{\epsilon_h^2}\alpha_1\log(T)\right)$$

while by Lemma C.3 and Lemma E.6 we have:

$$T_{h,2} \le \widetilde{\mathcal{O}}\left(\frac{1}{\epsilon_h}K^{m+1}n^{2m+2}\left(m^7(B + \log(T))\log\frac{1}{\delta'} + \binom{N+n}{m}\right)\right)$$

$$= \widetilde{\mathcal{O}}\left(\frac{1}{\epsilon_h}K^{m+1}n^{2m+2}\log(T)\left(m^7(B + \log(T))\log\frac{1}{\delta} + \binom{N+n}{m}\right)\right)$$

$$= \widetilde{\mathcal{O}}\left(\frac{1}{\epsilon_h}K^{m+1}n^{2m+2}\log^2(T)\left(m^7 B \log\frac{1}{\delta} + \binom{N+n}{m}\right)\right)$$

$$= \widetilde{\mathcal{O}}\left(\frac{1}{\epsilon_h}\alpha_2\log^2(T)\right).$$

Now we proceed to upper bound the regret of the algorithm. We let $R_h$ the regret accumulated during epoch $h$. First, we consider the single epoch $h = 1$, where the regret at each round is at most one. We can thus bound $R_1$ as:

$$R_1 \le T_{1,1} + T_{1,2}$$

$$\le \widetilde{\mathcal{O}}\left(\frac{1}{\epsilon_1^2}\alpha_1\log(T) + \frac{1}{\epsilon_1}\alpha_2\log^2(T)\right)$$

$$= \widetilde{\mathcal{O}}\left(K^2\alpha_1\log(T) + K\alpha_2\log^2(T)\right).$$

Consider now an epoch $2 < h < H$, and recall that $H \le \log_4(5T) = 1/2\log(5T)$ by Lemma 4.6. By Lemma D.5 applied to the previous epoch, the regret of each round during this epoch is at most $14K\epsilon_{h-1} = 28K\epsilon_h$, as $\epsilon_h = \frac{1}{K2^{h-1}}$. Therefore:

$$\sum_{h=2}^{H} R_h \le \mathcal{O}\left(K\epsilon_h(T_{h,1} + T_{h,2})\right)$$

$$\le \widetilde{\mathcal{O}}\left(\sum_{h=2}^{H} K\epsilon_h\left(\frac{1}{\epsilon_h^2}\alpha_1 + \frac{1}{\epsilon_h}\alpha_2\right)\log^2(T)\right)$$

$$= \widetilde{\mathcal{O}}\left(\sum_{h=2}^{H} K\left(\frac{1}{\epsilon_h}\alpha_1 + \alpha_2\right)\log^2(T)\right)$$

$$= \widetilde{\mathcal{O}}\left(K^2\log^2(T)\alpha_1\sum_{h=2}^{H} 2^{h-1} + \alpha_2\log^2(T)\right)$$

$$\le \widetilde{\mathcal{O}}\left(\log^2(T)K^2\alpha_1 2^H + \alpha_2\log^2(T)\right)$$

$$= \widetilde{\mathcal{O}}\left(K^2\alpha_1\sqrt{T} + \alpha_2\log^2(T)\right).$$

Putting all together, with probability at least $1 - \delta$ the regret of Algorithm 1 is:

$$R_T \le \widetilde{\mathcal{O}}\left(K^2\alpha_1\log(T) + K\alpha_2\log^2(T) + K^2\alpha_1\sqrt{T} + \alpha_2\log^2(T)\right)$$

$$= \widetilde{\mathcal{O}}(K^2 \alpha_1 \sqrt{T} + K \alpha_2 \log^2(T)).$$

The proof is concluded by observing that $\binom{N+n}{m} \leq (N+n)^m$ and performing simple computations. $\qquad\square$

## F. Technical Lemmas

**Lemma F.1.** *Let $\mathcal{P} \subseteq \Delta(\mathcal{A}_{\mathrm{L}})$ be a polytope and $x_1, x_2 \in \mathcal{P}$. Let also $u : \mathcal{P} \to [0,1]$ be an affine linear function, with $u(x_1) \leq u(x_2)$. Then for every $y \in [u(x_1), u(x_2)]$ there exists some $x_y \in \mathcal{P}$ belonging to the segment between $x_1$ and $x_2$ such that $u(x_y) = y$. When $u(x_1) = u(x_2)$, then $u(x') = u(x_1)$ for every $x'$ in the segment between $x_1$ and $x_2$.*

*Proof.* Suppose $u(x_1) < u(x_2)$ and consider a generic point $x_\lambda$ belonging to the segment between $x_1$ and $x_2$ and parametrized by $\lambda \in [0,1]$ as:

$$x_\lambda := x_2 + (x_1 - x_2)\lambda.$$

This point has utility $u(x_\lambda) = y$ when:

$$u(x_\lambda) = u(x_2) + \lambda(u(x_1) - u(x_2)) = y,$$

that is:

$$\lambda = \frac{y - u(x_2)}{u(x_1) - u(x_2)}.$$

It is easy to see that when $y \in [u(x_1), u(x_2)]$, $\lambda$ belongs to $[0,1]$. By convexity, it also holds that $x_\lambda \in \mathcal{P}$.

Suppose now $u(x_1) = u(x_2)$. Then $u(x_\lambda) = u(x_1) + \lambda(u(x_1) - u(x_1)) = u(x_1)$ for every $\lambda \in [0,1]$, concluding the proof. $\qquad\square$

**Lemma F.2.** *Let $\mathcal{P} \subseteq \Delta(\mathcal{A}_{\mathrm{L}})$ be a polytope with $\mathrm{vol}(\mathcal{P}) > 0$, and let $x \in \mathrm{interior}(\mathcal{P})$, where volume and interior are relative to the hyperplane containing $\Delta(\mathcal{A}_{\mathrm{L}})$. Then, if $x \in \mathcal{H}$ for some half-space $\mathcal{H}$, it holds that $\mathrm{vol}(\mathcal{P} \cap \mathcal{H}) > 0$.*

*Proof.* Let $H_\Delta$ be the hyperplane containing $\Delta(\mathcal{A}_{\mathrm{L}})$. We observe that if $H_\Delta \subseteq \mathcal{H}$, then the statement is trivially satisfied, as $\mathcal{P} \cap \mathcal{H} = \mathcal{P}$ has non-zero volume. We thus suppose that $H_\Delta \not\subseteq \mathcal{H}$. Since $x \in \mathrm{interior}(\mathcal{P})$, there exists some sphere

$$\mathcal{B}_\epsilon(x) = \{x' \in \mathbb{R}^m \mid \|x' - x\|_2 \leq \epsilon\}$$

of radius $\epsilon > 0$ such that $\mathcal{B}_\epsilon(x) \cap H_\Delta \subseteq \mathcal{P}$. There must exist some point $x^\circ$ at distance $0 < \epsilon_{\mathrm{small}} \leq \epsilon$ from $x$ that belongs to both $H_\Delta$ and $\mathrm{interior}(\mathcal{H})$. Now take any $x' \in \mathbb{R}^m$ such that $\|x^\circ - x'\|_2 \leq \epsilon^\circ$, with $0 < \epsilon^\circ < \epsilon - \epsilon_{\mathrm{small}}$, then:

$$\|x' - x\|_2 \leq \|x' - x^\circ\|_2 + \|x^\circ - x\|_2 \leq \epsilon^\circ + \epsilon_{\mathrm{small}} \leq \epsilon.$$

Therefore, we have $\mathcal{B}_{\epsilon^\circ}(x^\circ) \subseteq \mathcal{B}_\epsilon(x)$. As a result:

$$\mathcal{B}_{\epsilon^\circ}(x^\circ) \cap H_\Delta \subseteq \mathcal{B}_\epsilon(x) \cap H_\Delta \subseteq \mathcal{P}.$$

It follows that $x^\circ \in \mathrm{interior}(\mathcal{P})$. To conclude the proof, we observe that since $x^\circ \in \mathrm{interior}(\mathcal{H})$, $\mathrm{interior}(\mathcal{P} \cap \mathcal{H})$ is non-empty. $\qquad\square$

