# OpenReview forum: "Learning in Bayesian Stackelberg Games With Unknown Follower's Types"
_ICML.cc/2026/Conference — ICML 2026 regular_

### Official Review · Reviewer_7qNJ · 2026-03-09

**Soundness:** 4
**Presentation:** 3
**Significance:** 3
**Originality:** 4
**Overall Recommendation:** 4
**Confidence:** 3

**Summary:**

This paper investigates online learning in Bayesian Stackelberg Games (BSGs) under a highly uninformative setting where the leader completely lacks prior knowledge of the follower's type, payoff functions, or the type distribution. The authors establish a strong negative result: under standard *action feedback* (observing only the follower's best response), any learning algorithm suffers regret exponential in the bit-complexity of the follower's payoffs. Conversely, under *type feedback* (the follower's type is revealed at the end of the round), the authors propose a modular multi-epoch learning algorithm that achieves a sublinear regret of $O(\sqrt{T})$, albeit with an exponential dependence on the number of leader actions.

**Compliance With Llm Reviewing Policy:**

Affirmed.

**Final Justification:**

Based on the authors' rebuttal, I am raising my score to Weak Accept.

The authors fully address my primary concern regarding the seemingly contradictory type feedback assumption. The rebuttal's concrete example clarifies how the Follower’s type can be revealed without exposing the underlying payoff structure. My conceptual doubts are fully resolved, and I agree my previous citations do not apply to this scenario.

Overall, online learning in Bayesian Stackelberg games without knowing the follower’s payoffs is a novel and realistic problem. Theoretical results, including both a negative impossibility result under *action feedback* and a positive sublinear regret algorithm under *type feedback*, are sound. I appreciate that high-level descriptions of the algorithms and the intuition for the proofs are adequately presented. However, I believe that more discussions about concrete, real-world application examples (e.g. examples mentioned by the authors in Rebuttal) are needed to help readers better understand the paper.

**Key Questions For Authors:**

1. Can the authors provide further justification regarding type feedback? In the motivating security game example (Page 1), the authors state that it is unreasonable to assume the defender knows the target preferences of attackers. Then how does the defender perfectly observe the attacker's *type* after a single interaction without knowing their payoffs?

2. This framework essentially involves the leader actively querying the follower to deduce their utility function via best responses. How does this connect to *Active Inverse Reinforcement Learning* [2] or *Revealed Preference Theory* [3], where an agent/seller similarly learns from an environment/buyer that only provides best responses but not their types?

[2] Lindner, D., Krause, A., & Ramponi, G. (2022). Active exploration for inverse reinforcement learning. Advances in Neural Information Processing Systems, 35, 5843-5853.

[3] Roth, A., Ullman, J., & Wu, Z. S. (2016, June). Watch and learn: Optimizing from revealed preferences feedback. In Proceedings of the forty-eighth annual ACM symposium on Theory of Computing (pp. 949-962).

**Limitations:**

Yes.

**Strengths And Weaknesses:**

**Strengths:**

- **Solid Theoretical Results:** The work provides strong theoretical results, including both a negative impossibility result under *action feedback* and a positive sublinear regret algorithm under *type feedback*. The technical analysis seems sound.

- **Modular Algorithm Design:** The proposed learning algorithm is conceptually structured and modular, which makes the theoretical analysis tractable and logically sound.

**Weakness:**

- **Contradictory Assumptions:** The core motivation of the paper is that assuming knowledge of the follower's payoffs is unrealistic. However, the proposed *type feedback* setting requires the leader to observe the follower's type after each round. In classical Bayesian games formalized by Harsanyi [1], a player's type inherently contains *all* their private information, including their payoff function. It is conceptually unclear how the leader can perfectly identify a type identifier without knowing the underlying payoff structure it represents.

- **Lack of Empirical Evaluation:** The paper currently contains no empirical evaluation, making it difficult to assess the practical feasibility of the proposed algorithm. Further, an evaluation example would help clarify the interpretation of the assumptions.

- **Organization:**  The presentation is technically heavy. Moving some lemmas and proofs to the appendix, increasing more high level intuitions and running examples would help improve readability.

[1] Harsanyi, J. C. (1967). Games with incomplete information played by “Bayesian” players, I–III Part I. The basic model. Management science, 14(3), 159-182.

---

> ### Author Rebuttal · Authors · 2026-03-31
>
> We thank the Reveiwer for the comments. We will surely improve the presentation in the final version of the paper, adding examples and pictures to describe the evolution of the search space $\mathcal{X}_h$ across differet epochs.
>
> > Can the authors provide further justification regarding type feedback? In the motivating security game example (Page 1), the authors state that it is unreasonable to assume the defender knows the target preferences of attackers. Then how does the defender perfectly observe the attacker's type after a single interaction without knowing their payoffs?
>
> Let us clarify a possible misunderstanding in our definition of "type". In our setting, we follow a modern computer-science approach where a type $\theta \in \Theta$ is just an identifier (one can think of a natural number from $1$ to $K$), see, e.g. Letchford et al. (2009). This type is associated with the corresponding utility function $u^{F}_\theta$, which is not the type itself.
>
> Our type-feedback scenario allows to observe the identifier $\theta$, but not the corresponding $u^F_\theta$ .
>
>
>
> As a more practical example, in security games, the defender knows in advance all the possible types of attackers that they may encounter (e.g., in airport security, there might be unauthorised intruders, insider threats, terrorists, or smugglers), and a security guard can clearly distinguish the type of a criminal they find. However, they do not know in advance which are their preferences over possible targets, which must be learned through repeated interactions. In other settings, such as Web platforms, the possible follower's "type identifiers" could be user categories, which are clearly observable by the platform (leader).
>
> >This framework essentially involves the leader actively querying the follower to deduce their utility function via best responses. How does this connect to Active Inverse Reinforcement Learning [2] or Revealed Preference Theory [3], where an agent/seller similarly learns from an environment/buyer that only provides best responses but not their types?
>
> The papers pointed out by the Reviewer are indeed relevant for the literature on learning in Stackelberg games in general, but their models and techniques are substantially far away from those in our paper. Crucially, our model is a classical Bayesian Stackelberg game in normal form, where the leader's utility function is inherently piece-wise linear, and thus it is neither convex/concave nor continuous over the leader's strategy space. Instead, the papers pointed out by the Reviewer study models where the reward/utility to be optimised is somehow continuous and convex/concave. Crucially, the discontinuities of our model make the techniques in those papers inapplicable, and thus we need a completely different approach. If the Reviewer feels that a comparison with those papers is important, we will add it in the camera ready version of the paper.

---

> > ### Author Rebuttal · Reviewer_7qNJ · 2026-04-04
> >
> > Thanks for the authors’ detailed responses. I admit that my previous understanding of ‘type’ was incorrect. The concrete example provided in the rebuttal successfully resolved my confusion. Given this clarification, I agree that the papers I previously cited are not relevant. I will raise my score accordingly.

---

### Official Review · Reviewer_5FBL · 2026-03-11

**Soundness:** 4
**Presentation:** 3
**Significance:** 4
**Originality:** 3
**Overall Recommendation:** 5
**Confidence:** 4

**Summary:**

The authors study the learning problem of a leader in a Bayesian Stackelberg game setting. Specifically, the leader interacts in an online fashion with a sequence of followers, each of which has an unknown type that defines their utility function and that is drawn from an underlying distribution over a finite set of types. At each round the leader commits to a strategy in the simplex over her pure actions and a follower type is drawn from the underlying distribution. The follower then observes the strategy of the leader and best responds to it. The leader’s goal is to maximize her expected utility. Crucially however, the authors assume that the leader does not know the distribution from which the types are drawn or the actual utility function of each follower type. This significantly extends previous works on learning in Bayesian Stackelberg games, in which the utilities of the follower types are assumed to be known to the leader.

The authors study two settings, the action feedback model, where the leader observes only the best-response of the follower at the end of each round, and the type feedback model, in which the leader gets to observe both the best-response and the type of follower at the end of each round. For the action feedback model, the authors prove a negative result, showing that any learning algorithm must suffer regret that scales linearly with the number of rounds for a number of rounds that is exponential to the bit complexity of a follower’s payoffs. This yields any possible regret guarantees in this setting essentially impractical.

For the type feedback model, the authors propose an algorithm that works in phases, where each phase has 3 components that respectively aim to: (1) estimate the distribution and the set of most likely follower types to appear, (2) for each followers’ action profile learn the subset of the leader’s strategy space that induces the corresponding best-response and is approximately optimal, and (3) refine the leader’s strategy space. The authors prove O(\sqrt{T}) regret guarantee when the number of leader actions is fixed, which is tight given lower bounds from previous works.

**Compliance With Llm Reviewing Policy:**

Affirmed.

**Final Justification:**

I keep my prior very positive assessment, which the rebuttal reinforced. I find the results of the paper to be sound and the theoretical development very interesting and highly non-trivial. Minor weaknesses (mostly in presentation), I believe can be easily addressed.

**Key Questions For Authors:**

I am curious about the computational complexity of your approach. Can you please comment on this?

**Limitations:**

yes

**Strengths And Weaknesses:**

The authors study an important problem in online learning for Bayesian Stackelberg games, where the leader does not observe the types of the followers she interacts with. This makes the setting both challenging from a technical standpoint and highly relevant for practical applications.

The submission appears technically sound, and the theoretical development interesting. The paper is also well organized overall. That said, it is fairly notation-heavy, which can make it difficult to follow at times. For the next version, I would encourage the authors to include a table summarizing the most frequently used notation.

The algorithmic approach, which is the paper’s main technical contribution, is quite intuitive, although the analysis requires significant work and it’s highly non-trivial. In addition, in light of prior work, the regret bound is tight, which yields a complete characterization of the optimal regret in this setting.

Overall, I find the paper’s results important and valuable for both the learning theory and Stackelberg games communities, and therefore I vote for acceptance.

---

> ### Author Rebuttal · Authors · 2026-03-31
>
> We thank the Reviewer for the insightful comments and for positively evluating our work.
>
> > The submission appears technically sound, and the theoretical development interesting. The paper is also well organized overall. That said, it is fairly notation-heavy, which can make it difficult to follow at times. For the next version, I would encourage the authors to include a table summarizing the most frequently used notation.
>
> We thank the reviewer  for the suggestion. We will add the table in the final version of the paper.
>
> > I am curious about the computational complexity of your approach. Can you please comment on this?
>
> We thank the Reviewer for pointing out this aspect, and we will certainly include an additional discussion of the computational aspects in the final version of the paper. Our approach can be implemented with per-round running time polynomial in $K$ and $n$ when $m$ is constant (i.e., small), an assumption that is already required to avoid exponential regret.
>
> We provided a clear pseudocode that is simple to understand and analyze theoretically, but an efficient implementation exploits two keys observations. Specifically, there are only at most $\mathcal{O}(K^m n^{2m})$ non-empty regions $\mathcal{X}_h(a_h)$, and each one is specified by $\mathcal{O}(Kn)$ hyperplanes. This observation allows to implement the algorithm with polynomial time and space complexity. One needs only to keep track of the non-empty regions, and to iterate over them when prescribed by the pseudocode.
>
> The exponential running time when $m$ is not constant is expected, as the offline problem of computing an optimal commitment in a Bayesian Stackelberg game is NP-hard in such a case. Indeed, the main bottleneck of our algorithm is the computation of $\underline{\text{OPT}}_h$ in Algorithm 4, which requires solving at most $\mathcal{O}(K^m n^{2m})$ LPs. This is the same computational cost required to solve the offline problem.

---

> > ### Author Rebuttal · Reviewer_5FBL · 2026-04-02
> >
> > Thank you for your response!

---

### Official Review · Reviewer_fQ4P · 2026-03-13

**Soundness:** 2
**Presentation:** 3
**Significance:** 3
**Originality:** 2
**Overall Recommendation:** 4
**Confidence:** 4

**Summary:**

This paper investigates the online learning problem in Bayesian Stackelberg Games with unknown follower types. Unlike previous literature that generally assumes the leader knows the payoff matrices of all possible types, this paper considers then setting where the leader has absolutely no prior knowledge of either the payoff matrices or the probability distribution of the follower's types. The paper shows under pure action feedback, any algorithm inevitably suffers an exponential regret lower bound of $\Omega(2^{2L})$ with respect to the bit complexity of the payoffs. The authors then propose an algorithm under the type feedback setting, which successfully achieves a sublinear regret bound of $\tilde{\mathcal{O}}(\sqrt{T})$.

**Compliance With Llm Reviewing Policy:**

Affirmed.

**Final Justification:**

I maintain my weak accept recommendation. The paper addresses a theoretically important setting in Bayesian Stackelberg games, and I find the overall technical development strong. The rebuttal also addressed my core concerns.

**Key Questions For Authors:**

Questions:

- Regarding the lower bound construction, is there a more generalized insight? For instance, if we impose structural assumptions on the utility functions (e.g., a constant margin/gap between actions, or Lipschitz continuity), is there any hope/insight for finding a low-regret learning algorithm under pure action feedback?
- Could you explicitly provide the computational complexity of running Algorithm 3 and Algorithm 4 per epoch? How does the time complexity scale with the number of leader actions $m$, types $K$, and follower actions $n$?
- The hidden constants in the regret upper bound depend exponentially on the leader's action dimension $m$, which the authors argue is unavoidable. Could you provide a clear, intuitive explanation as to why this exponential dependence is a fundamental information-theoretic limit?
- Can you provide 1-2 concrete, real-world examples where a leader perfectly observes the follower's type post-interaction but has zero prior knowledge of the payoff matrices associated with those types?

**Limitations:**

Yes

**Strengths And Weaknesses:**

Strengths:

- This paper is the first to solve the regret minimization problem in Bayesian Stackelberg Games under the setting where the leader is completely ignorant of the follower's payoffs. This elegantly relaxes the unrealistic "known payoff matrices" assumption prevalent in prior BSG learning literature, significantly advancing the theoretical frontier of this field.
- The paper is exceptionally well-structured and provides a complete theoretical loop. The proofs are logically sound and thorough.

Weaknesses:

- The negative result (the $\Omega(2^{2L})$ regret lower bound under action feedback) seems to rely on an extremely adversarial construction where the utility difference between actions is exponentially small in the bit complexity $L$. While theoretically sound for a worst-case bit-complexity bound, this argument becomes less practically meaningful for large $T$ if we assume a constant suboptimality gap between discrete actions. It remains unclear whether this impossibility result holds in more benign, generalized settings.
- The paper establishes a egret bound of $\tilde{\mathcal{O}}(\sqrt{T})$, but doesn't fully discuss the computational complexity of the proposed method. Specifically, in the Find-Partition subroutine (Algorithm 3) and the Prune step (Algorithm 4), the algorithm requires computing and maintaining the intersections of multiple $m$-dimensional polytopes ($\mathcal{P}_\theta(a)$) across $K$ types. Given that high-dimensional polytope intersection can scale exponentially, this raises severe concerns about the practical feasibility of the algorithm.
- The core problem setting assumes the leader is entirely ignorant of the utility functions but simultaneously receives perfect, exact feedback about the follower's private type after each round. This feels somewhat artificially engineered to bridge the gap between existing non-Bayesian and Bayesian Stackelberg literature. The paper lacks a compelling real-world scenario to justify this specific information structure.

---

> ### Author Rebuttal · Authors · 2026-03-31
>
> We thank the Reviewer for the insightful comments.
>
>
> > Regarding the lower bound construction, is there a more generalized insight? For instance, if we impose structural assumptions on the utility functions (e.g., a constant margin/gap between actions, or Lipschitz continuity), is there any hope/insight for finding a low-regret learning algorithm under pure action feedback?
>
> First, let us remark that the literature on online learning Stackelberg games is focused on instance-independent (worst case) guarantees (see, e.g., Letchford et al. (2009); Peng et al. (2019); Bacchiocchi et al. (2025)). In these settings, it is customary to provide lower bounds on difficult instances. Furthermore, our lower bound can also be restated as follows: for any $T$, there is an instance represented by $\Theta(\log(T))$ bits where the algorithm attains $\Theta(T)$ regret. This is a standard formulation, as even the classical $\Omega(\sqrt{T})$ lower bound in multi-armed bandits requires $\Theta(\log(T))$ bits to encode the expected rewards of the arms. We make the dependency on $L$ explicit because in our setting the prior knowledge of this parameter is required.
>
> To overcome this lower bound we should thus introduce strong assumptions. Regarding Lipschitz continuity, the fact that the leader’s expected utility is only piecewise linear (and not continuous) is the main challenge of the problem. We observe however that it is piecewise constant in the lower bound instance, with a “suboptimality gap” of 1 between diferent best-response regions. The follower's utility is instead continuos over the leader's strategy space.
>
>
>
> > Could you explicitly provide the computational complexity of running Algorithm 3 and Algorithm 4 per epoch? How does the time complexity scale with the number of leader actions $m$, types $K$, and follower actions $n$?
>
> Both algorithms can be implemented with per-round running time polynomial in $K$ and $n$ when $m$ is constant, an assumption that is already required to avoid exponential regret. One needs to take into account that there are only at most $\mathcal{O}(K^m n^{2m})$ non-empty regions $\mathcal{X}_h(a_h)$. Iterations over action-profiles can thus be implemented as iterations over these non-empty regions. Furthermore, each one of these regions can be represented by at most $\mathcal{O}(Kn)$ hyperplanes, ensuring that the memory space is also polynomial.
>
> The exponential running time when $m$ is not constant is unavoidable, as the offline problem of computing an optimal commitment in a Bayesian Stackelberg games is NP-hard in such a case. Indeed, the most expensive rounds from a computational perspective are those where Algorithm 4 is executed. This algorithm solves at most $\mathcal{O}(K^m n^{2m})$ LPs to compute $\underline{\text{OPT}}_h$, which is the same computational cost required to solve the offline problem. Let us also observe that polytopes are naturally stored in their $H$-representation, which allows to compute their intersections efficiently.
>
> >The hidden constants in the regret upper bound depend exponentially on the leader's action dimension $m$, which the authors argue is unavoidable. Could you provide a clear, intuitive explanation as to why this exponential dependence is a fundamental information-theoretic limit?
>
> This lower bound is due to Peng et al. (2019). Intutively, even with a single type, it is necessary to enumerate and query all the vertices of the best-response regions (exponential in $m$ in the worst case) to ensure that they have been learned correctly and that an optimal commitment has been found. Otherwise, the learner may miss a region that provides a very large utility.
>
> >Can you provide 1-2 concrete, real-world examples where a leader perfectly observes the follower's type post-interaction but has zero prior knowledge of the payoff matrices associated with those types?
>
> In most of the real-world applications of interest, it is reasonable to assume that the leader knows a set of possible follower's "type identifiers" and has a way to distinguish among them. Instead, it is quite unreasonable that the leader has complete knowledge of the exact utilities of each of these types. For instance, in security games, the defender knows in advance all the possible types of attacker that they may encounter (e.g., in airport security, there might be unauthorised intruders, insider threats, terrorists, or smugglers), but they do not know in advance which are their preferences over possible targets, which must be learned through repeated interactions. In other settings, such as Web platforms, the possible follower's "type indentifiers" could be user categories, which are clearly observable by the platform (leader).

---

> > ### Author Rebuttal · Reviewer_fQ4P · 2026-04-03
> >
> > Thank you for the rebuttal. I will maintain my score.

---

### Official Review · Reviewer_Gyhs · 2026-03-16

**Soundness:** 3
**Presentation:** 3
**Significance:** 3
**Originality:** 3
**Overall Recommendation:** 4
**Confidence:** 3

**Summary:**

This paper studies online learning in Bayesian Stackelberg Games (BSGs) where both the type and the utility function of the follower are unknown. The authors first show a negative result, demonstrating a regret that scales exponentially in the number of bits used to encode the utility of the follower if only the action of the follower is revealed to the leader. To this end, the authors consider an easier setting where both the realized type and the adopted action of the follower are revealed to the leader. In this setting, the proposed algorithm achieves an $O(\sqrt{T})$ regret.

**Compliance With Llm Reviewing Policy:**

Affirmed.

**Final Justification:**

All my previous concerns are resolved.

**Key Questions For Authors:**

Please see my question above.

**Limitations:**

Please see my question above.

**Strengths And Weaknesses:**

**Strengths**
1. **Presentation:** Most parts of this paper are well written.
2. **Motivation and Significance:** Previous online learning works on BSGs assume prior knowledge (e.g., the utility function), which might be restrictive in practice. This work is the first to study online learning in BSGs without requiring knowledge of the utility function and the probability distribution of the follower’s type. The problem is well motivated. Also, I think the theoretical results established in this work are valuable to the community.

**Weaknesses**

I think it would be better if some parts of this work could be further explained or discussed. For instance:
1. It would be better if the concrete form of $\beta$ in Theorem 4.7 could be given in the main body of the paper. The current statement “$\beta= \dots$ when $m$ is constant” is not very clear.
2. Could the authors provide some discussion on the tightness of the dependence on $K$ in Theorem 4.7?

**Additional Questions**:
I am interested in the adversarial setting, where the distribution of the follower’s type can change adversarially. Do the authors think that sublinear regret in this setting can be obtained or not?

---

> ### Author Rebuttal · Authors · 2026-03-31
>
> We thank the Reviewer for the insightful comments.
>
> >It would be better if the concrete form of $\beta$ in Theorem 4.7 could be given in the main body of the paper. The current statement “$\beta = \dots$ when $m$ is constant” is not very clear.
>
> We agree with the Reviewer, we will use the additional space in the final version of the paper to explicitly state the value of $\beta$ in the main theorem.
>
>
> >Could the authors provide some discussion on the tightness of the dependence on $K$ in Theorem 4.7?
>
> We do not have a tight lower bound with respect to the number of types $K$ when $m$ is constant in this setting. We observe that for large $T$, the leading term in the regret bound of our algorithm is $K^2\sqrt{T}$. We conjecture that a refined analysis could improve this dependence, but this remains an open problem.
>
> > I am interested in the adversarial setting, where the distribution of the follower’s type can change adversarially. Do the authors think that sublinear regret in this setting can be obtained or not?
>
> The adversarial setting is very interesting scenario and a natural future research direction. However, we cannot really guess whether sublinear regret can be attained in such a setting, as it would require substantially different techniques. Our approach heavily relies on estimating the unknown distribution $\mu$, and fails when the sequence of types is chosen adversarially. A setting where the utility of each type is known would be a good starting point, as positive results have been obtained under other models with commitments under such assumptions (see, e.g., [1]).
>
> [1] Castiglioni, M., Celli, A., Marchesi, A., & Gatti, N. (2020). Online bayesian persuasion. Advances in neural information processing systems, 33, 16188-16198.

---

> > ### Author Rebuttal · Reviewer_Gyhs · 2026-04-03
> >
> > I thank the authors for their responses, and I have no further questions.

---

### Decision · Program_Chairs · 2026-04-30

**Decision:**

Accept (regular)

**Comment:**

The reviewers agree that the problem addressed in this paper is important, the results obtained are sound, and the theoretical development is interesting. In the initial reviews, there were concerns about the presentation and the validity of certain assumptions, but these appear to have been adequately addressed through the rebuttal.

Therefore, I recommend acceptance of this paper.

As the reviewers have commented, I recommend revisions such as improving the presentation and adding descriptions of concrete real-world applications.